# Loss of p53 activates thyroid hormone via type 2 deiodinase and enhances DNA damage

Annarita Nappi [1], Caterina Miro[1], Antonio Pezone [2], Alfonso Tramontano[3], Emery Di Cicco[1], Serena Sagliocchi[1], Annunziata Gaetana Cicatiello[1], Melania Murolo[1], Sepehr Torabinejad [1], Elena Abbotto [4], Giuseppina Caiazzo[1], Maddalena Raia[5], Mariano Stornaiuolo[6], Dario Antonini [2], Gabriella Fabbrocini[1], Domenico Salvatore [5,7], Vittorio Enrico Avvedimento [8] & Monica Dentice[1,5] ✉

The Thyroid Hormone (TH) activating enzyme, type 2 Deiodinase (D2), is functionally required to elevate the TH concentration during cancer progression to advanced stages. However, the mechanisms regulating D2 expression in cancer still remain poorly understood. Here, we show that the cell stress sensor and tumor suppressor p53 silences D2 expression, thereby lowering the intracellular THs availability. Conversely, even partial loss of p53 elevates D2/TH resulting in stimulation and increased fitness of tumor cells by boosting a significant transcriptional program leading to modulation of genes involved in DNA damage and repair and redox signaling. In vivo genetic deletion of D2 significantly reduces cancer progression and suggests that targeting THs may represent a general tool reducing invasiveness in p53-mutated neoplasms.

Thyroid hormones (THs, T4 and T3) are key endocrine regulators of cellular growth, differentiation, and metabolism[1]. Many physiological effects of THs are mediated mainly by their canonical action via TH nuclear receptors (TRs, encoded by *THRA* and *THRB* genes), resulting in induction and/or inhibition of target genes transcription[1–4]. Besides the capacity of the thyroid gland to produce the normal amount of THs for circulation, T3 and T4 concentrations in peripheral tissues are modulated by the combined actions of a family of three iodothyronine enzymes, the deiodinases D1, D2, and D3, that catalyze the initiation and termination of local TH action and finely control TH signaling in a time- and tissue-specific manner, regardless of serum hormone concentrations[1,5]. D2 catalyzes the T4-to-T3 conversion, thereby increasing the nuclear availability of the active hormone. Conversely, D3 is the main inactivator of THs, because it converts T4 and T3 into inactive molecules, rT3 and T2, respectively.

The last decades of research pointed to a critical role of THs in regulating the neoplastic process, affecting the formation and progression toward malignancy of various cancer types[1,6–8]. Research in animal models and in human patients has demonstrated that altered expression of the deiodinases and mutations or loss of function of *THRs* are common events in some tumors and can affect cancer cell proliferation, invasiveness, and angiogenesis[3,9–12]. Moreover, recent evidence suggested that hyperthyroidism increases the risk of several solid malignancies while hypothyroidism reduces the aggressiveness and delays the onset of cancer[13]. Although the role of D2 and D2-produced T3 in skin cancer is well established, the molecular network involved in the D2 expression regulation in epithelial tumorigenesis is still unknown. Based on the above consideration, we searched for upstream regulators of D2 expression in the context of Squamous Cell Carcinoma (SCC) and found that the tumor suppressor *Trp53* (mus musculus)/*TP53* (human), one of the most commonly mutated genes in over 50% of all human diagnosed cancers[14], acts as transcriptional inhibitor of D2. Here, we report that p53 physically binds to *Dio2* gene, encoding for the D2 protein, promoter, and suppresses D2 expression,

[1]Department of Clinical Medicine and Surgery, University of Naples "Federico II", 80131 Naples, Italy. [2]Department of Biology, University of Naples "Federico II", 80126 Naples, Italy. [3]Department of Precision Medicine, University of Campania "L. Vanvitelli", 80138 Naples, Italy. [4]Department of Experimental Medicine, University of Genoa, 16132 Genoa, Italy. [5]CEINGE, Biotecnologie Avanzate S.c.a.r.l., 80131 Naples, Italy. [6]Department of Pharmacy, University of Naples "Federico II", 80149 Naples, Italy. [7]Department of Public Health, University of Naples "Federico II", 80131 Naples, Italy. [8]Department of Molecular Medicine and Medical Biotechnology, University of Naples "Federico II", 80131 Naples, Italy. ✉e-mail: monica.dentice@unina.it

both at mRNA and protein levels, thus causing a reduction in intracellular active TH transactivation ability. Strikingly, we observed that D2 expression levels faithfully mirror the p53 status and function in vivo and in vitro.

Furthermore, in vivo ablation of D2 reduces the tumorigenesis dependent on the loss of p53, demonstrating a causative link between p53 depletion, the upregulation of D2 and the consequent evolution of cancer toward the invasive phenotype. To search for D2- and THs-dependent actions that can be a relevant part of the p53 signaling, we analyzed the RNA-seq data in tumors upon *Dio2* depletion and found upregulation of oxidative and DNA repair genes.

In this work, we reveal a key role of p53 in controlling TH action on metabolism, growth, and DNA Damage Response (DDR). The D2/TH axis is an important transcriptional network that, by repressing the expression of DNA repair and oxidative genes (pro-invasive, in many instances), favors the accumulation of genomic instability, metastasis, and resistance of epithelial cancers. Furthermore, these data suggest that targeting THs may represent a general tool to reduce the invasiveness of p53-mutated neoplasms.

## Results

### The expression of D2 is negatively regulated by p53 in SCC cancer cells

To identify the transcriptional regulators of *Dio2* gene, we performed in silico motif scanning analysis of the upstream *Dio2* promoter region (−1.2 kb to the Transcriptional Start Site, TSS). This region contains several binding sites for transcription factors, including 15 different decameric half-sites, with a consensus motif RRRCWWGYYY, recognized by the tumor suppressor p53 (Fig. 1a and Supplementary Fig. 1). We focused our studies on p53, based on its key role as transcriptional regulator of tumorigenesis, cell cycle, and DNA damage. Among the predicted p53 binding sites, 2 putative consensus motifs, hereinafter referred to as distal (P1: − 1224 to −1248 bp, likelihood score 0.916) and proximal (P2: −293 to −317 bp, likelihood score 0.944), showed the highest score for the functional and evolutionary cross-species conservation properties (Supplementary Fig. 1). Furthermore, the analysis of gene expression profiles and of the mutational signatures from the GEO Dataset GSE42677 (https://www.ncbi.nlm.nih.gov/geo/query/acc.cgi?acc=GSE42677) of human cutaneous SCCs[15,16] showed a striking inverse correlation between the *TP53* and its modulator, *ATM*, and *DIO2* expression in skin cancer progression (Pearson's correlation coefficient, ρ: $ρ_{ATM}$ = 0.86; $ρ_{TP53}$ = 0.95; $ρ_{DIO2}$ = −0.93) (Fig. 1b, c). To explore the possibility that p53 may be directly involved in the regulation of *DIO2* gene expression, SCC cancer cells were transfected with a wild-type p53 expression vector and Real-Time RT-qPCR analysis revealed that the exogenous expression of the wild-type p53 strongly suppresses D2 mRNA levels (Fig. 1d). Consistent with these data, the *Dio2* promoter fused with a LUC reporter (*Dio2*-LUC-WT) was significantly inhibited by p53 (Fig. 1e), along with a reduction of the intracellular TH action as demonstrated by the co-transfection of p53 with an artificial T3-responsive promoter, the TRE3-TK-LUC (Fig. 1f). Accordingly, D2 protein levels in primary keratinocytes from p53 null mice were markedly upregulated compared to D2 levels in keratinocytes derived from wild-type mice (Fig. 1g).

To confirm the role of p53 as a negative transcriptional regulator of D2, p53 was knocked down by the transfection of selective siRNAs. Indeed, p53 silencing restored the D2 mRNA expression, the *Dio2* promoter activity and induced the intracellular TH action (Fig. 1h–j). Furthermore, Chromatin Immuno-Precipitation (ChIP) assay demonstrated that p53 physically binds the *Dio2* promoter (Fig. 1k). Next, we generated *Dio2*-LUC deletion mutants to disrupt the P1 and P2 p53 binding sites selectively and to evaluate the D2 regulation. Transient transfection assays revealed that deletion of P1 (*Dio2*-LUC-Δ1 and Δ2) was sufficient to abrogate the p53-dependent D2 inhibition (Fig. 1l). Conversely, the deletion of P2 (*Dio2*-LUC-Δ3) did not influence the p53-

dependent inhibition of D2 activity (Fig. 1l), thus demonstrating that D2 responsiveness to p53 requires the physical interaction of P1 half-site in the *Dio2* promoter.

To further verify the role of P1 in D2 expression regulation, the core motif of P1 was mutagenized, generating the *Dio2*-LUC-mut promoter (*Dio2*-LUC-WT −nnnTGACAAGTCTnnn− versus *Dio2*-LUC-mut −nnnTGACCCCTCTnnn−) (Fig. 1m). As expected, the mutant promoter was not inhibited by p53. The luciferase activity was higher and insensitive to exogenous expressed p53 (Fig. 1m). Finally, we evaluated if the D2 inhibition by p53 occurred also in non-tumorigenic cells such human keratinocytes HaCaT. Overexpression of p53 in these cells downregulated D2 mRNA (Supplementary Fig. 2a)[17].

Altogether, these data demonstrate a close correlation between D2 expression levels and nuclear p53 activity, and both were connected with the TH intracellular levels.

### Loss of p53 sustains D2 expression

To gain a better insight into the mechanism(s) linking p53 to D2 expression, we performed structural-functional studies of p53 inactivation. Four plasmids carrying p53 DNA-binding domain mutations (R248W, E258K, P278F, and E286K) were ectopically expressed in SCC011 cells (Fig. 2a). All the p53 mutants lost the p53-dependent D2 inhibition (Fig. 2b), did not inhibit the *Dio2* promoter activity (Fig. 2c) and reduced the intracellular TH action (Fig. 2d), thus confirming that mutations in the p53 DNA-binding domain lead to a complete abrogation of p53-dependent D2 inhibition. Next, we analyzed the cellular response to UV-C-radiation (a Single-Strand-Breaks, SSBs-inducing agent)[18] of two SCC cancer cell lines with different endogenous p53 configurations, respectively SCC011[19] and SCC13 (p53 mutant)[20] (Fig. 2e). To this end, the two SCC cell lines were treated with a single dose of UV-C-radiation (100 μJ/cm²) and analyzed at different time points (0, 1, 2, 4, and 8 h) after the exposure.

As expected, in SCC011 cells, D2 mRNA expression was reduced after UV-C-treatment (Fig. 2f) along with the induction of known p53 target genes, as demonstrated by a significant induction of p21 and the pro-apoptotic factor BAX, and the concomitant reduction of Cyclin-D1, MDM2 and the anti-apoptotic factor BCL-2 (Supplementary Fig. 2b). By contrast, the UV-C-treatment in the p53 mutant SCC13 cells did not modify the expression of D2 and even resulted in a slight but not statistically significant D2 upregulation (Fig. 2g). The downregulation of D2 in SCC011 cells paralleled the rapid and transient activation of p53 after UV-C-radiation by Ser-15 phosphorylation (Fig. 2h and Supplementary Fig. 2c), and by the concomitant autophosphorylation of the ATM on Ser-1981 (Fig. 2h and Supplementary Fig. 2c). Similar results were observed in HaCaT cells in which, p53 activation upon UV-C-treatment reduced D2 expression, consistently with an induction of p21 (Supplementary Fig. 2d).

Notably, using a different UV-lamp that produces UV-A and UV-B radiations, which better reproduce the physiological exposure to the sun lights, we confirmed that, upon UV-A/B induced p53 phosphorylation, the expression of D2 was significantly reduced in SCC011 cells and did not change in SCC13 cells (Fig. 2i–l and Supplementary Fig. 2e, f).

Furthermore, pharmacological modulation of the ATM-p53 axis showed that etoposide, which induces DNA double-strand-breaks, DSBs, and activates p53[21], downregulated D2 in p53-dependent manner, while KuDOS (that inhibits ATM/p53 signaling)[22] induced D2 (Fig. 2m–o).

These results confirm the p53-dependent D2 inhibition and suggest a protective role of p53 against D2 activation.

Having established that p53 is an up-stream regulator of D2 expression, we wondered if p53 also regulates the expression of D1, which activates the pro-hormone T4 to T3 and D3, which inactivates the THs, since both D1 and D3 promoter regions contain several p53 binding sites (Supplementary Fig. 3a, b). D3 mRNA increased in

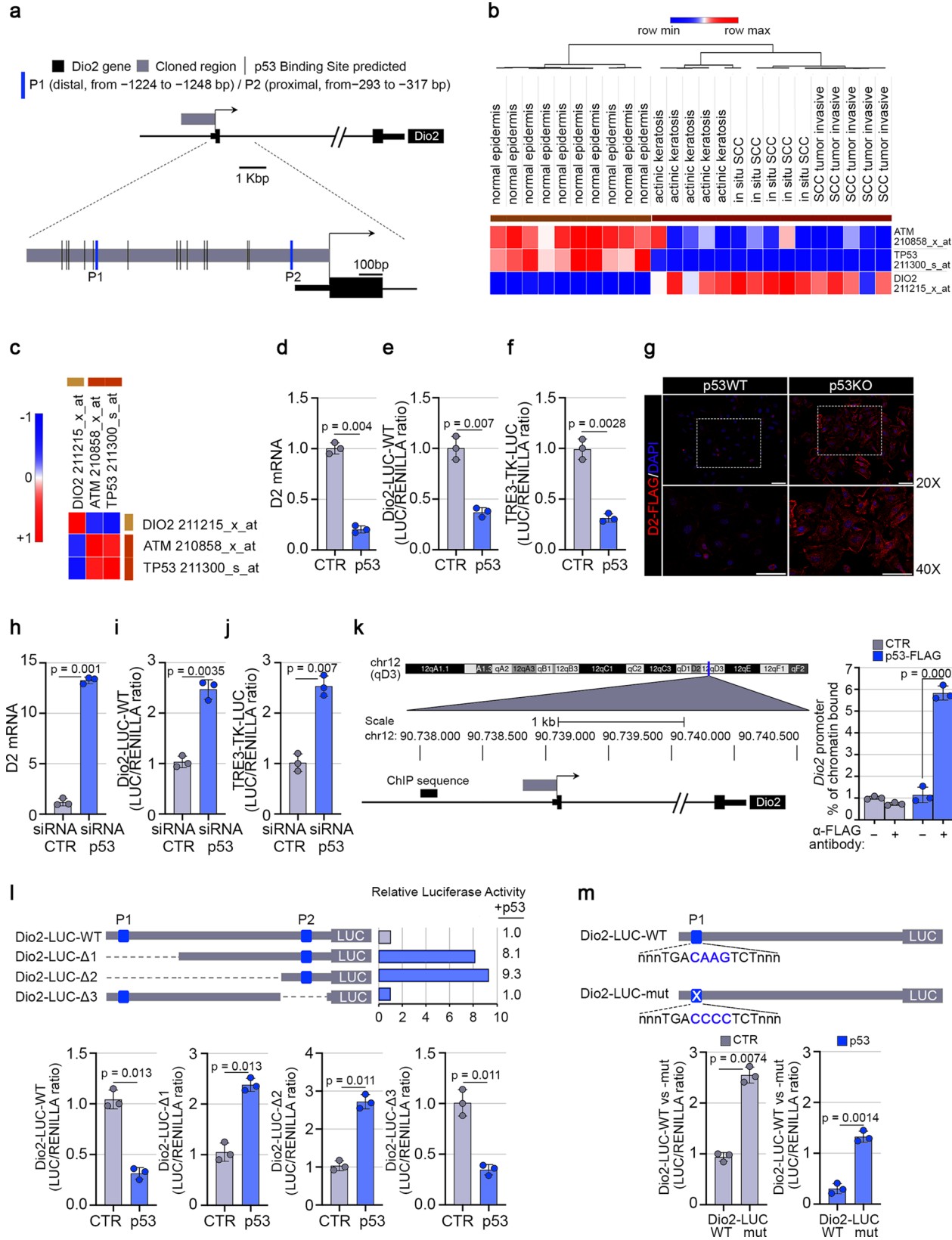

SCC011 cells transfected with wild-type p53, while the expression of the mutated p53 protein did not modify D3 mRNA levels (Supplementary Fig. 3c). Similarly, D3 mRNA levels were upregulated in UV-C exposed cells (Supplementary Fig. 3d). Conversely, D1 mRNA levels were undetectable in SCC cells and in p53-transfected cells. We conclude that p53 expression in SSC cells inhibits TH levels by stimulating

D3 (which inhibits THs) and repressing D2 (which activates THs), resulting in a strong inhibition of the THs-mediated cancer progression.

To further investigate if the effects of THs on SCC cells are only mediated by the genomic pathway of the T3/TRs complexes, we also analyzed the activation of the non-genomic pathway of THs that

**Fig. 1 | p53 downregulates the Type 2 Deiodinase transcription in SCC cancer cells. a** Schematic localization of 15 different putative p53 consensus motifs within the *Dio2* promoter region. Blue bars indicate the distal (P1: − 1224 to −1248 bp) and the proximal (P2: − 293 to −317 bp, score 0.944) p53 consensus motifs. **b** Expression heatmap of *ATM, TP53,* and *DIO2* genes, differentially expressed across normal epidermis, actinic keratosis, in situ, and invasive SCC samples from GSE42677. **c** Pearson's correlation analysis was performed for the same dataset samples as in (**b**), by using Morpheus heatmap visualization software (Pearson's correlation coefficient, ρ: $\rho_{ATM}$ = 0.86; $\rho_{TP53}$ = 0.95; $\rho_{DIO2}$ = −0.93). **d** D2 mRNA expression was measured by Real-Time PCR in SCC011 cells transiently transfected with a p53-expressing vector or a control empty plasmid (CTR) (*n* = 3 independent experiments). **e, f** Transient co-transfection of a Dio2-LUC-WT promoter (**e**) and an artificial T3-responsive promoter, TRE3-TK-LUC (**f**), with a p53-expressing vector or a CTR plasmid, in SCC011 cells (*n* = 3 independent experiments). **g** D2-FLAG expression levels were measured by immunofluorescence (IF) analysis of mouse primary keratinocytes from D2-Flag/p53 null mice and D2-Flag/p53 wild-type mice transfected with the p53 or the CTR plasmid (representative of 3 images per sample, *n* = 3 mice). Data are presented as overviews (top rows) and higher magnification (bottom rows). Magnification 20X and 40X. Scale bars represent 50 µm. **h** D2

mRNA expression was measured by Real-Time PCR in SCC011 cells transiently transfected with a cocktail of siRNAs used against the endogenous p53 messenger (*n* = 3 independent experiments). **i, j** Transient co-transfection of a cocktail of siRNAs used against the endogenous p53 messenger along with the Dio2-LUC-WT promoter (**i**) and the TRE3-TK-LUC promoter (**j**), in SCC011 cells (*n* = 3 independent experiments). **k** Chromatin Immuno-Precipitation (ChIP) assay was performed in SCC011 cells transiently transfected with a p53-FLAG expressing vector, followed by Real-Time PCR with primers specific for *DIO2* promoter region proximal to TSS. Graph shows the Real-Time PCR results with % chromatin bound (*n* = 3 independent experiments). **l** Schematic representation of the Dio2-LUC deletion mutants (Dio2-LUC-Δ1, Δ2, and Δ3) for the analysis of the selective disruption of the P1 or the P2 p53 binding sites within the *Dio2* promoter region (*n* = 3 independent experiments). **m** Schematic representation of the point mutation of the P1 p53 binding site for the generation of the Dio2-LUC-mut promoter and evaluation of the *Dio2* mutant promoter activity *versus Dio2* wild-type promoter activity in SCC011 cells (*n* = 3 independent experiments). All the results in (**d**–**f**); (**h**–**m**) are shown as means ± SD from at least 3 separate experiments. *p*-values were determined by two-tailed Student's t-test. *$p < 0.05$, **$p < 0.01$, ***$p < 0.001$. Source data are provided as a Source data file.

involve the αVβ3-Integrin-mediated activation of the phospho-ERK/MAPK axis[4,23]. Interestingly, in our experimental condition, the levels of phospho-ERK/ERK1/2 and the expression of p38-MAPK did not change upon THs treatment, thus suggesting that, in these conditions and in these cells, THs act mainly by modulation of transcription (Supplementary Fig. 4).

### Genetic D2 depletion attenuates the p53-dependent tumorigenesis

To investigate the functional role of p53-mediated D2 regulation in the skin tumor progression, we performed in vivo studies by manipulating p53-dependent expression of D2 in mice in which SCC tumors were generated using the two-step chemically induced carcinogenesis model[24]. To this end, we generated transgenic mice with heterozygous *Trp53* gene deletion and homozygous *Dio2* genetic ablation in the epidermal compartment. Thus, we crossed p53+/−;D2fl/fl mice[25,26] with K14-CreERT mice[27]. First, we depleted *Dio2* by treating (i) K14-CreERT−/−;p53+/+/D2fl/fl (CTR or p53WT;D2WT), (ii) K14-CreERT−/−;p53+/−;D2fl/fl (p53KO+/−;D2WT), and (iii) K14-CreERT+/−;p53+/−;D2fl/fl (p53KO+/−;D2KO) mice with Tamoxifen and applied the mutagen DMBA to their dorsal skin one week later (Fig. 3a). Effective D2 depletion from the epidermal compartment was confirmed by PCR, as previously described by Miro et al.[10] (Supplementary Fig. 5a).

12 weeks after DMBA treatment, analyses of morphological features showed that, compared to the CTR mice, p53KO+/−;D2WT had a higher number and larger size of tumors (Fig. 3b–e), with p53KO+/−;D2WT mice developing skin papillomas at an average of 4−6 weeks after the first application of DMBA, *versus* 10−12 weeks in p53KO+/−;D2KO mice. Notably, p53KO+/−;D2KO mice formed fewer and smaller tumors compared to p53KO+/−;D2WT mice, thus indicating that, while p53 depletion enhances the frequency (number and size of tumors at each time point) and the incidence (time of lesion appearance) of tumors in the D2 wild-type background, loss of D2, on the other hand, drastically reduces the p53-depletion-dependent tumorigenesis (Fig. 3b–e). In addition, survival analysis on a number of 12 mice for each genotype revealed that heterozygous *Trp53* gene deletion in the D2 wild-type background (p53KO+/−;D2WT) shortens the Overall-Survival (OS) in SCC tumorigenesis as analyzed by Kaplan-Meier (*p* = 0.036, Fig. 3c). Consistently with the in vitro data, the expression of D2 was strongly upregulated in p53KO+/−;D2WT compared to CTR mice (Fig. 3f), and this further reinforces the notion that D2 downregulation by p53 is an essential barrier to epidermal tumorigenesis. This is also shown by the pronounced tumor growth in p53KO+/−;D2WT mice associated with higher tumor progression and invasive conversion. Indeed, as shown in Fig. 3g−j, in p53KO+/−;D2WT tumors, we

observed a significant reduction of epithelial markers (i.e., E-Cadherin, Krt6 and Krt14, Fig. 3g, j and Supplementary Fig. 5b), associated with a remarkable upregulation of EMT markers (i.e., N-Cadherin, Vimentin, Zeb-1, Snail and Twist, Fig. 3h, j and Supplementary Fig. 5b) and a higher Keratin-8 (Krt8) expression. The same results were obtained by Hematoxylin/Eosin (H&E) staining (Fig. 3k) and immunofluorescence (IF) analysis of epithelial and mesenchymal markers in p53KO+/−;D2WT *versus* p53KO+/−;D2KO mice (Fig. 3l and Supplementary Fig. 5c).

Together, these data demonstrate that genetic D2 depletion reduces the SCC tumor formation in p53-depleted cells, highlighting the lower propensity of these tumors to acquire an invasive phenotype.

### D2 depletion reduces metastasis formation in p53+/− tumors

To monitor the spread of cancer to distant sites, we measured the circulating tumor cells (CTCs) in the peripheral blood of p53KO+/−;D2WT and p53KO+/−;D2KO mice. A total of 12 mice blood samples for each genotype were analyzed in this study (Fig. 4a). Blood samples were collected and analyzed for the expression of three specific markers, i.e., Krt8[28], Krt19[29], and c-Met[30], clinical biomarkers for skin cancer patients. Based on the fold change of expression, it was evident that *Krt8*, *Krt19,* and *c-Met* in p53KO+/−/D2WT mice were expressed at higher levels than p53KO+/−;D2KO (Fig. 4b−d).

To further investigate whether the D2 depletion influences the p53-dependent metastasis formation, we isolated the inguinal lymph nodes from p53KO+/−;D2WT and p53KO+/−;D2KO mice and performed histopathologic examination with H&E (Fig. 4e) and IF analysis (Fig. 4f) for the expression of Krt8 and CXCR4 as markers of metastasis[31,32]. Lymph nodes from p53KO+/−;D2WT mice were larger and showed higher expression of Krt8 and CXCR4 compared to p53KO+/−;D2KO (Fig. 4f). These data are corroborated by the analysis of the TCGA dataset, showing that high D2 expression is associated with poor survival (Supplementary Fig. 6) and that D2 and p53 levels were inversely correlated in different tumor types (Supplementary Figs. 7−9).

Therefore, D2/TH depletion or inhibition represents a valid strategy to reduce tumor invasiveness and metastasis in tumors with p53 mutations.

### D2/TH inhibit the expression of DNA damage repair genes in p53+/− cells

The data presented above show that D2 is a key component of the p53 depletion-dependent tumorigenic program in SCC. Since the only known function of D2 is the activation of THs, we reasoned that the nuclear action of TH might be involved in such a process. Thus, we

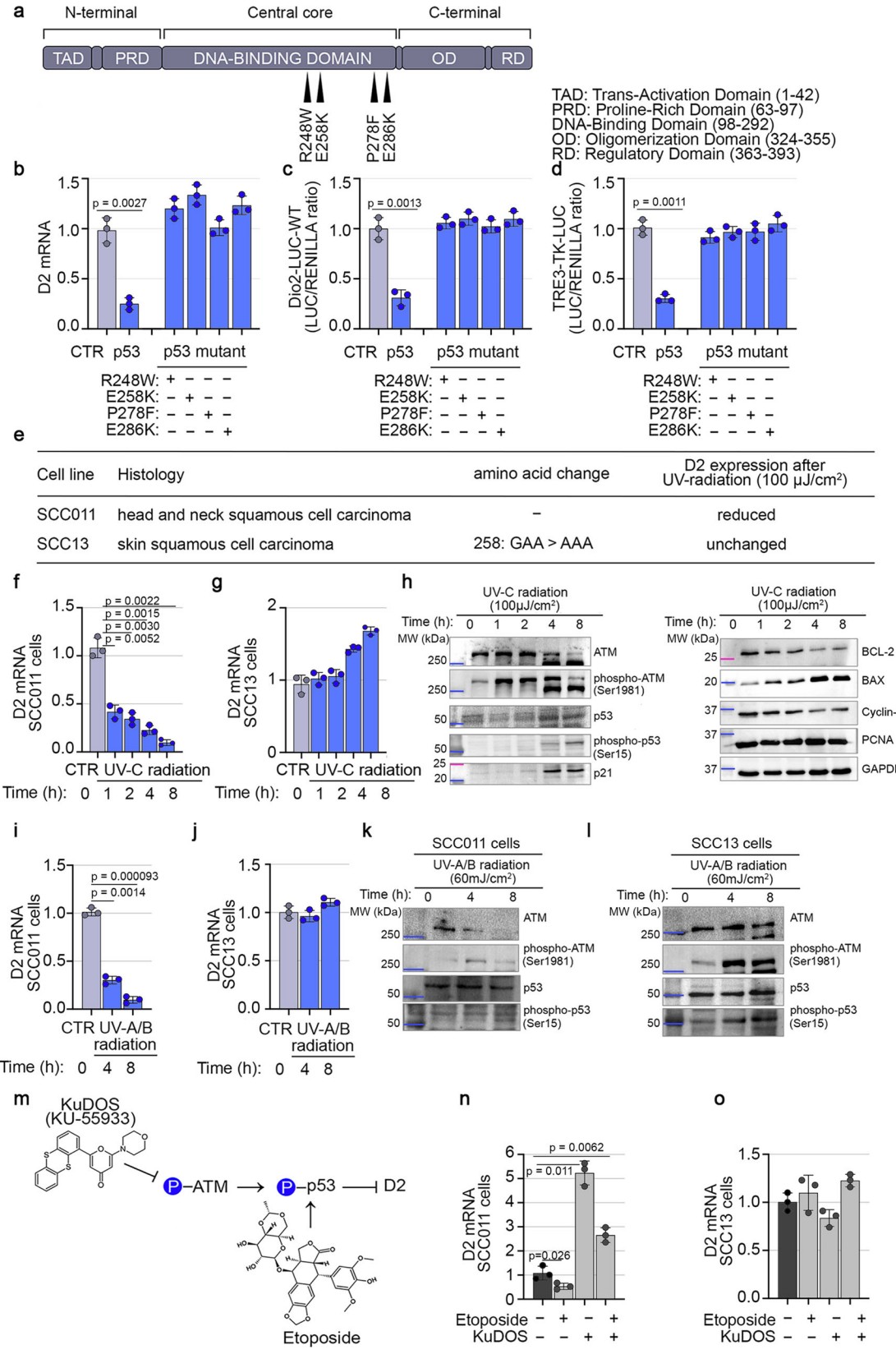

analyzed the genes regulated by epidermal-specific THs inactivation in the context of *Trp53* heterozygous gene deletion. We performed RNA-seq analysis (TruSeq mRNA-Seq library kit, Illumina) following Keratin-14 (Krt14)-guided Laser Capture Microdissection (LCM) on the epithelial compartment of SCC tumors isolated from dorsal skin of p53KO[+/−];D2KO *versus* p53KO[+/−];D2WT mice (*n* = 3 mice/group)

(Fig. 5a). We found 111 Differentially Expressed Genes (DEGs, 65 of them negatively regulated for value of Log$_2$ Fold-Change (FC) ≤ −1.5 and 46 positively regulated for value of Log$_2$ Fold-Change (FC) ≥ −1.5, both with *p* < 0.05) between p53KO[+/−];D2KO and p53KO[+/−];D2WT mice (Fig. 5b, c, Supplementary Fig. 10 and Supplementary Data 1). Based on Gene Ontology (GO) enrichment analysis in Biological Processes (BP)

**Fig. 2 | Loss of p53 abrogates D2 inhibition. a** Schematic representation of the p53 protein structural domains and localization of four distinct p53 DNA-contact mutations (R248W, E258K, P278F and E286K). **b** D2 mRNA expression was measured by Real-Time PCR in SCC011 cells transiently transfected with four DNA-contact mutant p53 plasmids, carrying different p53 DNA-mutations (R248W, E258K, P278F and E286K) ($n = 3$ independent experiments). **c, d** Transient co-transfection of a Dio2-LUC-WT promoter (**c**) and TRE3-TK-LUC (**d**), with four distinct mutant p53 plasmids, in SCC011 cells ($n = 3$ independent experiments). **e** Characteristics of two human SCC cancer cell lines with different endogenous p53 status, SCC011 (p53 wild-type status) and SCC13 (p53 mutant status). **f, g** D2 mRNA expression was measured by Real-Time PCR in SCC011 (**f**) and SCC13 (**g**) cells, in a time course response to DNA damage induced by exposure to UV-C radiation (100 μJ/cm²) at the indicated time points (0, 1, 2, 4 and 8 h) ($n = 3$ independent experiments). **h** Western Blot analysis of cell cycle (Cyclin-D1), apoptosis (BCL-2, BAX) and regulatory proteins (ATM, phospho-ATM Ser-1981, p53, phospho-p53 Ser-15, p21) in SCC011 cells following exposure to UV-C radiation. Cell lysates were analyzed at the indicated time points (0, 1, 2, 4, and 8 h) after treatment. A

representative result of 3 independent experiments is shown. GAPDH and PCNA were used as internal markers. **i, j** D2 mRNA expression was measured by Real-Time PCR in SCC011 (**i**) and SCC13 (**j**) cells, in a time course response to DNA damage induced by exposure to UV-A/B radiation (30 mJ/cm²) at the indicated time points (0, 4 and 8 h) ($n = 3$ independent experiments). **k, l** Western Blot analysis of regulatory proteins (ATM, phospho-ATM Ser-1981, p53, phospho-p53 Ser-15) in SCC011 (**k**) and SCC13 (**l**) cells following exposure to UV-A/B radiation. Cell lysates were analyzed at the indicated time points (0, 4, and 8 h) after treatment. A representative result of 3 independent experiments is shown. **m** Schematic representation of mechanism of action of two antitumor drugs, KuDOS (KU-55933) and etoposide, and their effects on the activation of ATM serine/threonine kinase and p53 protein. **n, o** D2 mRNA expression was measured by Real-Time PCR in SCC011 (**n**) and SCC13 (**o**) cells, treated with etoposide (50.0 μM/30 min) and KuDOS (100.0 μM/1 h) as single drugs or used in combination ($n = 3$ independent experiments). All the results are shown as means ± SD from at least 3 separate experiments. $p$-values were determined by two-tailed Student's t-test. *$p < 0.05$, **$p < 0.01$, ***$p < 0.001$. Source data are provided as a Source data file.

(Supplementary Fig. 11a, b), we focused our attention on genes involved directly or indirectly in the redox signaling and DDR, as *Mad1l1* (FC = 10.912, $p = 0.04331$), *Pak1* (FC = 14.3639, $p = 0.02239$), *Pold3* (FC = 10.249, $p = 0.04921$), *Rnf111* (FC = 13.5701, $p = 0.01818$) and *Rnf169* (FC = 12.3197, $p = 0.03145$) (Supplementary Fig. 11c), hereinafter referred as DNA Repair Genes (DRGs). To validate the RNA-seq analysis, we measured the expression of the above-mentioned genes in D2WT and D2KO SCC cells. To this aim, we suppressed D2 expression in vitro in SCC011 cells (here referred as D2KO) using CRISPR/Cas9 technology and measured the expression of the identified DEGs compared to the parental SCC011 cells (here referred as D2WT). mRNA expression levels of *MAD1L1*, *PAK1*, *POLD3*, *RNF111*, and *RNF169* were significantly upregulated in SCC cells upon D2 depletion (Fig. 5d). Notably, the expression of the above-mentioned genes was restored when D2KO cells were treated with T3 (Supplementary Fig. 11d) and was inhibited when D2 expression was turned-on by doxycycline administration (Supplementary Fig. 11e). Moreover, the THs-dependency of the DEGs was further demonstrated in in vivo experiments in which wild-type mice, in which the SCC formation was induced by chemical carcinogenesis, were treated with THs for three weeks before the sacrifice. Under these conditions, the expression levels of *Mad1l1*, *Pak1*, *Pold3*, *Rnf111*, and *Rnf169* were significantly reduced compared to untreated mice (Supplementary Fig. 11f), thus confirming that TH negatively regulates the identified DEGs. Furthermore, to assess if these genes were directly regulated by THs via thyroid hormone responsive elements (TREs), we analyzed in silico the DNA region located upstream of the promoter comprising sequences from −2.0 kb relative to the TSS (Supplementary Figs. 12–14). The analysis revealed putative consensus TRE binding sites in all the genes analyzed: 12 consensus TRE binding sites in *MAD1L1* promoter region (Supplementary Fig. 12a), 22 consensus TRE binding sites in *PAK1* promoter region (Supplementary Fig. 12b), 15 consensus TRE binding sites in *POLD3* promoter region (Supplementary Fig. 13a), 11 consensus TRE binding sites in *RNF111* promoter region (Supplementary Fig. 13b) and 17 consensus TRE binding sites in *RNF169* promoter region (Supplementary Fig. 14a). ChIP analysis revealed the chromatin TRs binding site in *MAD1L1* (Fig. 5e, i) and *POLD3* promoters (Fig. 5e, ii), suggesting that these genes are directly repressed by THs (Fig. 5e) and not in the other promoters (Fig. 5e). However, we show that the expression of all these genes was dependent on p53 because they were induced by wild-type and not by p53 mutant in SCC cancer cells (Fig. 5f). Similar results were validated by in vivo experiments, in which we compared *Mad1l1*, *Pak1*, *Pold3*, *Rnf111*, and *Rnf169* gene expression in p53KO⁺/⁻;D2WT and p53KO⁺/⁻;D2KO *versus* CTR mice in which SCC was induced by chemical carcinogenesis (Fig. 5g). Finally, we confirmed that the p53 induction of the DEGs occurs also in the untransformed HaCaT cells, (Supplementary Fig. 14b). In conclusion, the expression of these DNA repair

genes is amplified in the wild-type p53 background, because p53, by repressing D2, downregulates THs, which inhibit directly and indirectly their expression.

To further corroborate the existence of an ATM/p53 axis that antagonizes the effects of THs, we analyzed the expression of D2 and the identified DEGs in the GEO Dataset GSE42677 including samples from Normal Skin (NS), Actinic Keratosis (AK), and SCC. This analysis confirms a significant negative correlation between *DIO2* and *TP53* and a positive correlation between *ATM*, *TP53*, and the identified THs-target DRGs (Pearson's correlation coefficient, ρ: $\rho_{DIO2} = -0.98$; $\rho_{MAD1L1} = 0.51$; $\rho_{PAK1} = 0.54$; $\rho_{ATM} = 0.73$; $\rho_{POLD3} = 0.80$; $\rho_{TP53} = 0.93$; $RNF111 = 0.23$) (Supplementary Fig. 14c,d).

These data show that D2 and D2-produced TH silence and repress *MAD1L1*, *PAK1*, *POLD3*, *RNF111*, and *RNF169* genes. Their inhibition leads to a faulty DDR, which favors genome instability and tumor progression.

### Enhancement of TH signal leads to increased DNA damage

The logical implication of the data shown above is that the progression of epidermal tumors is favored by upregulation of D2 and, ultimately, of TH, because the hormone alters the expression of relevant DNA repair genes. THs stimulate the expression of many genes involved in oxidative metabolism including mitochondrial genes[33,34]. Also, induction or repression of transcription in general[35] and/or transcription by nuclear hormones[36] triggers a genome-wide wave of G oxidation necessary to assemble the transcription pre-initiation complex and to prime repression or activation[36].

Confocal microscopy images of THs-exposed SCC011 cells show the nuclear and perinuclear (mitochondria) accumulation of the major DNA oxidation adduct, the 8-oxo-deoxyguanosine (8-oxo-dG)[37] (Fig. 6a and Supplementary Fig. 15a). Since 8-Oxo-dG is recognized by the OxoG glycosylase (OGG1)[38], we measured the recruitment of OGG1/2[38], at the promoter region of the five THs-target DRGs, *MAD1L1*, *PAK1*, *POLD3*, *RNF111*, and *RNF169*. ChIP analysis revealed that THs triggers the accumulation of OGG1/2 at *MAD1L1*, *POLD3*, *RNF111*, and *RNF169* promoter regions (Supplementary Fig. 15b). However, in the presence of excess or continuous TH levels, such as in SCC011 cells, the repression of these repair genes leads to defective or incomplete DNA repair as shown by the accumulation of double-strand breaks visible as γ-H2AX and RAD51-foci (Fig. 6b). To demonstrate that TH induces an inefficient DNA repair response we used a specific engineered cell system to quantify the repair of a DSB by homologous recombination (HR)[39]. This system enables the evaluation of DNA repair thanks to the expression of the Sce-I enzyme causing one DSB/genome in one GFP copy (cassette I), which can be repaired from the second copy (cassette II) by HR. Notably, FACS analysis revealed that THs treatment significantly reduced HR (Supplementary Fig. 15c). These results show

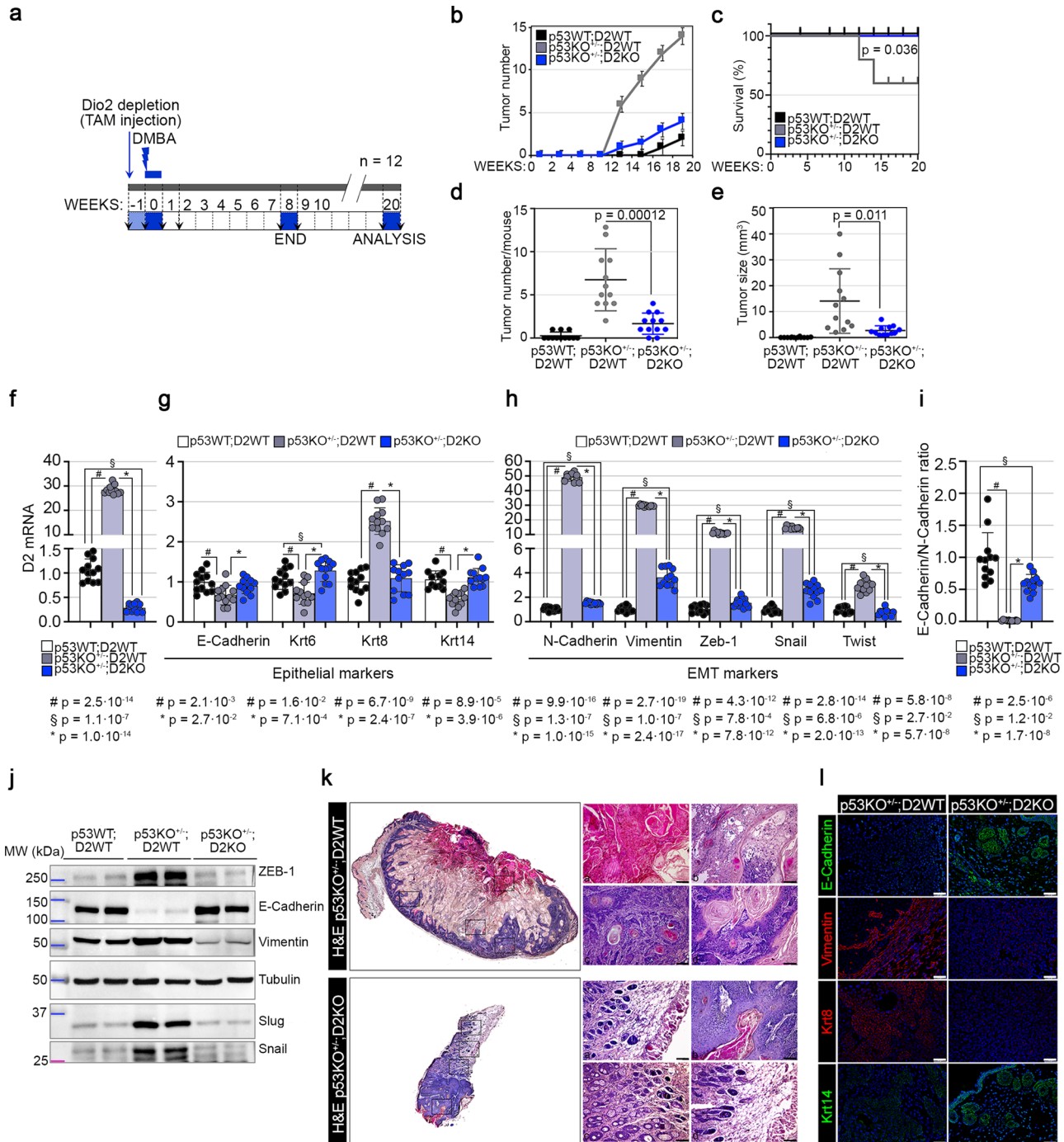

**Fig. 3 | Epidermal attenuation of TH signaling reduces the p53-dependent tumorigenesis. a** Schematic representation of the two-step chemically induced carcinogenesis experiments. **b**–**e** Tumor number (**b**), percentage of survival (**c**), tumor number/mouse (**d**), and tumor size (**e**) were analyzed during the two-step chemically induced carcinogenesis experiments in (i) p53WT;D2WT, (ii) p53KO[+/−]; D2WT and iii) p53KO[+/−];D2KO mice (n = 12 mice/group). **f** D2 mRNA expression was measured by Real-Time PCR in SCC tumors from dorsal skin of (i) p53WT;D2WT, (ii) p53KO[+/−];D2WT and iii) p53KO[+/−];D2KO mice (n = 12 mice/group). **g**–**i** mRNA expression of epithelial (**g**) and mesenchymal (**h**) markers and their ratio (**i**) was measured by Real-Time PCR in SCC tumors from dorsal skin of (i) p53WT;D2WT, (ii) p53KO[+/−];D2WT and (iii) p53KO[+/−];D2KO mice. **j**, Western Blot analysis of epithelial (E-Cadherin) and mesenchymal (Vimentin, ZEB-1, Snail, Slug) proteins in SCC

tumors from dorsal skin of (i) p53WT;D2WT, (ii) p53KO[+/−];D2WT and (iii) p53KO[+/−]; D2KO mice (n = 12 mice/group). A representative result of 3 independent experiments is shown. Tubulin was used as internal markers. **k** H&E of skin lesions from (i) p53KO[+/−];D2WT and (ii) p53KO[+/−];D2KO mice. Full skin-section reconstruction with Leica LAS X Navigator and representative zoom images. Magnification 20X. Scale bars represent 50 μm (representative of 3 images per sample, n = 3). **l** IF staining with epithelial (E-Cadherin, Krt14) and mesenchymal (Vimentin, Krt8) markers of skin lesions from (i) p53KO[+/−];D2WT and (ii) p53KO[+/−];D2KO mice (representative of 3 images per sample, n = 3 mice). Magnification 20X. Scale bars represent 50 μm. All the results are shown as means ± SD from at least 3 separate experiments. p-values were determined by two-tailed Student's t-test. *p < 0.05, **p < 0.01, ***p < 0.001. Source data are provided as a Source data file.

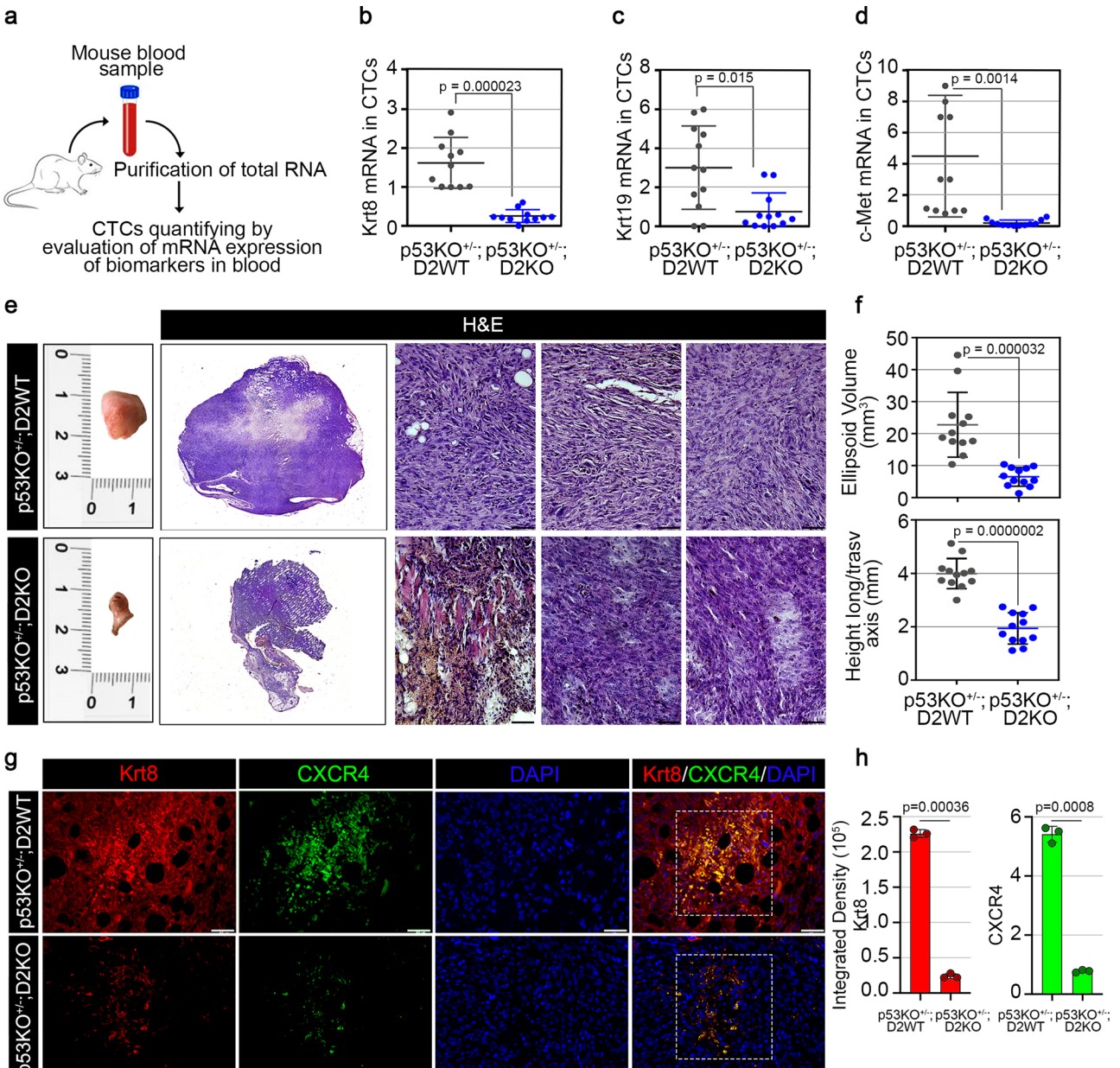

**Fig. 4 | Loss of D2 attenuates the p53-dependent tumor metastatization at distant sites. a** Graphical representation of the strategy used to isolate circulating tumor cells (CTCs) from mouse blood samples of DMBA-treated (i) p53KO[+/-];D2WT and (ii) p53KO[+/-];D2KO mice (*n* = 12 mice/group). **b–d** mRNA expression of three metastatic biomarkers, Krt8 (**b**), Krt19 (**c**), and c-Met (**d**), was measured by Real-Time PCR in CTCs isolated from blood samples of DMBA-treated (i) p53KO[+/-];D2WT and (ii) p53KO[+/-];D2KO mice (*n* = 12 mice/group). **e** Representative images of inguinal lymph nodes from (i) p53KO[+/-];D2WT and (ii) p53KO[+/-];D2KO mice treated with DMBA for 8 weeks and harvested at 20 weeks after *Dio2* genetic depletion and DMBA administration (*n* = 12 mice/group). H&E of a representative inguinal lymph node for each group of mice (representative of 3 images per sample, *n* = 3 mice).

Full lymph node-section reconstruction with Leica LAS X Navigator and representative zoom images. Magnification 20X. Scale bars represent 50 μm. **f** Ellipsoid volume (mm³) and height longitudinal/transversal axis (mm) of solid lymph nodes derived from (i) p53KO[+/-];D2WT and (ii) p53KO[+/-];D2KO mouse models (*n* = 12 mice/ group). **g** IF staining with Krt8 and CXCR4 markers of lymph nodes sections from (i) p53KO[+/-];D2WT and (ii) p53KO[+/-];D2KO mice. Magnification 20X. Scale bars represent 50 μm. **h** Relative quantification of Krt8 and CXCR4 immunofluorescent signals (Integrated Density) are represented by histograms (representative of 3 images per sample, *n* = 3 mice). All the results are shown as means ± SD from at least 3 separate experiments. *p*-values were determined by two-tailed Student's t-test. *p < 0.05, **p < 0.01, ***p < 0.001. Source data are provided as a Source data file.

that THs reduce the efficiency of the most specific and accurate type of repair (HR).

In order to confirm that D2 modulates the expression of genes involved in the DNA repair independently of p53 status, we performed experiments in SCC13 cells that stably carry an inducible Tet-ON-D2 construct, generated in our lab.[12] In these cells (p53 null) doxycycline administration turns on the D2 expression. As shown in Fig. 6c, d, the DOX-induced D2 increased the 8-oxo-dG and γ-H2AX/RAD51 levels.

Moreover, the DOX-induced D2 significantly reduced the expression of the DNA-repair genes (Supplementary Fig. 15d).

We controlled cell viability in basal condition and after the etoposide-induced DNA damage. THs reduced the fraction of viable cells (V) and increased early (E) and late (L) apoptotic and dead (N) cells (Fig. 7a), in basal condition and after etoposide treatment, contextually to the D2 and p21 regulation by etoposide (Supplementary Fig. 16a). Furthermore, we found an important and

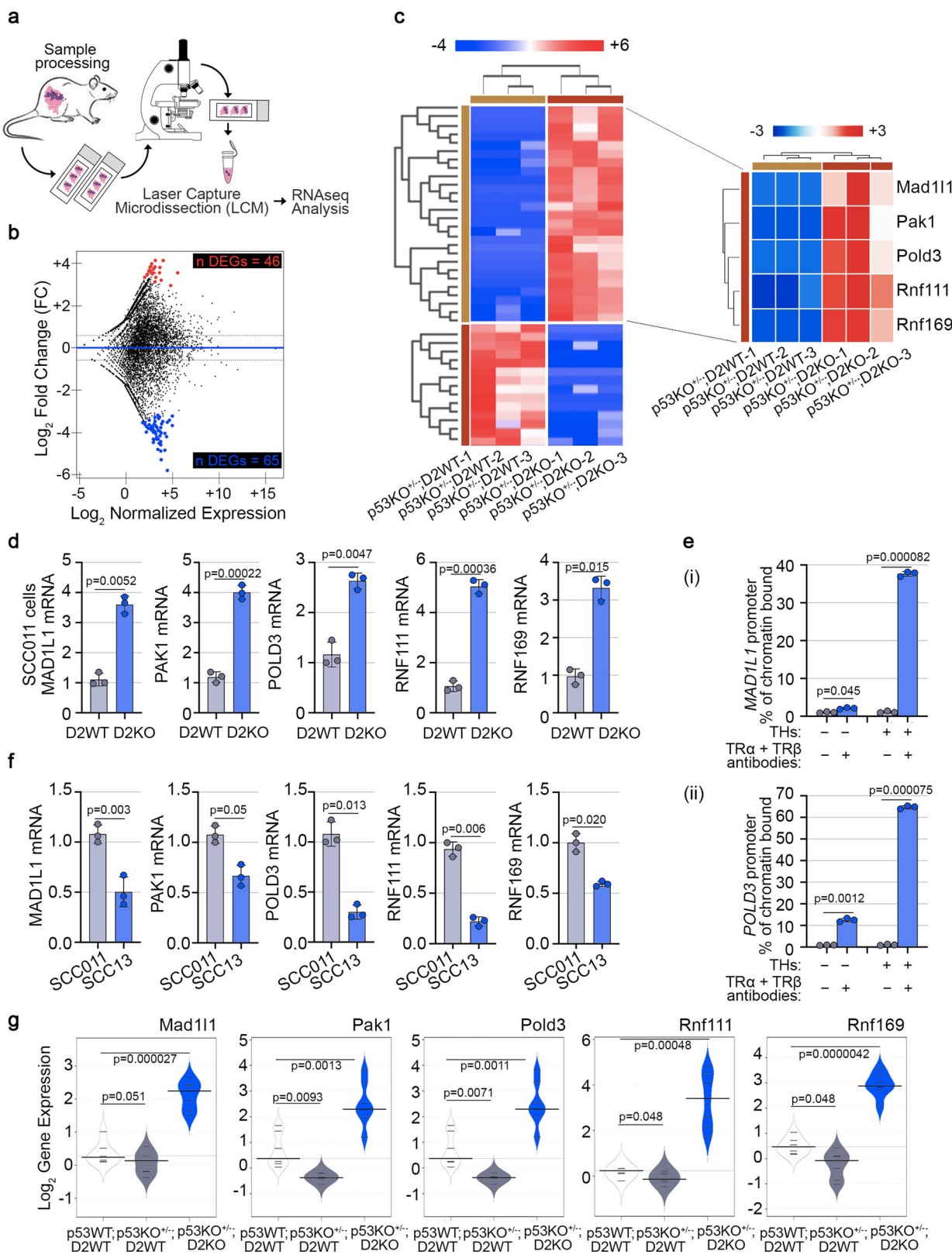

convincing result: when cultured in THs-deprived medium (Charcoal-Stripped Serum), etoposide-induced apoptosis was dramatically reduced, thus reinforcing our initial hypothesis that persistent TH levels are potent promoters of apoptosis (Fig. 7a). Similar results were obtained in another set of experiments, in which, after the UV-induced DNA damage in SCC011 cells, THs enhanced the percentage of apoptotic cells (Supplementary Fig. 16b). Moreover, THs-

dependent apoptosis was accompanied by a downregulation of all five DRGs, *MAD1L1*, *PAK1*, *POLD3*, *RNF111* and *RNF169* (Fig. 7b), in both cases, when cells were treated with THs alone or in combination with etoposide.

In conclusion, these data provide evidence for an important function of THs, which is intrinsically connected to the THs induction of oxidative metabolism. Cells with aberrant DNA damage are

**Fig. 5 | Epidermal-specific *Dio2* depletion fosters the DNA damage repair mechanisms in skin tumorigenesis. a** Graphical representation of the Laser Capture Microdissection (LCM)-strategy followed by RNA-seq analysis from skin tumor sections of (i) p53KO[+/−];D2WT and (ii) p53KO[+/−];D2KO mice. **b** MA plot shows differences in expression between (i) p53KO[+/−];D2WT and (ii) p53KO[+/−];D2KO mice. In black are represented genes that do not pass the parameters of the filters selected (−1.5 ≥ FC ≥ + 1.5, *p*-value ≤ 0.05). In red are represented genes upregulated in p53KO[+/−];D2KO *versus* p53KO[+/−];D2WT mice (FC ≥ + 1.5, *p*-value ≤ 0.05). In blue are represented genes downregulated in p53KO[+/−];D2KO *versus* p53KO[+/−];D2WT mice (FC ≤ − 1.5, *p*-value ≤ 0.05). **c** Expression heatmap of 111 Differentially Expressed Genes (DEGs) that were expressed across (i) p53KO[+/−];D2WT and (ii) p53KO[+/−];D2KO mice and zoom of the top 5 DEGs, namely *MAD1L1, PAK1, POLD3, RNF111,* and *RNF169,* selected from analysis. **d** mRNA expression of the top 5 DEGs

was measured by Real-Time PCR in SCC011 D2WT *versus* D2KO cells (*n* = 3 independent experiments). **e** ChIP of thyroid hormone receptors, TRα and TRβ, binding to the *MAD1L1* (i) and *POLD3* (ii) promoter was performed in SCC011 cells treated or not with THs (T3, 30.0 nM + T4, 30.0 nM) for 24 h. Graph shows the Real-Time PCR results with % chromatin bound as indicated (*n* = 3 independent experiments). **f** mRNA expression of the top 5 DEGs was measured by Real-Time PCR in SCC011 *versus* SCC13 cells (*n* = 3 independent experiments). **g** Bean Plots represent the Log₂ gene expression of the top 5 DEGs measured by Real-Time PCR in SCC tumors from dorsal skin of (i) p53WT;D2WT, (ii) p53KO[+/−];D2WT and (iii) p53KO[+/−];D2KO mice (*n* = 7 mice/group). All the results are shown as means ± SD from at least 3 separate experiments. *p*-values were determined by two-tailed Student's t-test. \**p* < 0.05, \*\**p* < 0.01, \*\*\**p* < 0.001. Source data are provided as a Source data file.

frequently eliminated by p53 and its partial loss amplifies DNA instability induced by THs.

## Discussion

The oncosuppressor p53 is either lost or mutated in about half of all human cancers[40], and the consequent p53 LOF has a critical impact on DNA damage repair, cell cycle checkpoint control, and apoptosis, which in turn result in the enhanced propensity of tumor cells to evolve toward invasive and metastatic transformation. Acting as transcriptional regulator of many different target genes, p53 regulates the cell fate following DNA damage.

Here, we report that among the p53-target genes, the *Dio2* gene, which encodes for the TH-activating enzyme, the Type 2 Deiodinase, is inhibited by p53. Indeed, both, p53 silencing in SCC cells and overexpression of different p53 mutated constructs result in upregulated/de-repressed D2 expression (Fig. 2). Importantly, the mutation in the same amino acid (E286 to-G or to-K), which is located in the DBD of p53, results in D2 upregulation, thus strengthening the concept that loss of p53 DNA-binding is responsible for the induction of D2. Moreover, D2 expression inversely correlates with the number of p53 wild-type alleles, since D2 expression in heterozygous p53 skin is already reduced when compared to wild-type skin and is then totally reduced in p53[-/-] skin. Notably, we also found that p53 positively regulates the expression of D3, suggesting a role for p53 as a potent modulator of the D2/D3 balance in cancer cells.

The p53-dependent negative regulation of D2 results in enhanced TH action in a p53 null context and explains a recent emerging notion that THs promote the progression of cancer toward malignancy and that tumors expressing high D2 levels are characterized by lower percent survival and higher recurrence[10]. However, although the correlation between high D2 expression and a low percent of survival can be found in different tumor types and this is a common feature of most of the human malignancies, it is not ubiquitously and constantly present during the whole tumor progression stages.

Although D2 has been associated with different hallmarks of cancer as enhanced EMT, migration and invasion of cancer cells, and angiogenesis promotion[10–12,41], the direct regulation by p53 raises the question of whether D2 can be involved in the DNA damage repair and in the genome stability.

We performed UV-radiation experiments as a tool for inducing DNA damage with a substantial proportion of DS breaks and p53 activation. We found that, when exposed to UV-radiation, SCC cells drastically reduce D2 expression in a p53-dependent manner. When exposed to the three types of UV light, SCC cells activate p53 by inducing its phosphorylation in Ser-15 as expected[42] and by upregulating p21 with a coupled ATM activation, which sense and organize DDR (Fig. 2h).

This suggests that the down-modulation of THs is consequent to DNA damage. Accordingly, analysis of RNA-seq data in D2WT *versus* D2KO SCC tumors indicated that loss of D2 leads to the upregulation of a set of genes involved in the DNA damage repair as of *Mad1l1, Pak1,*

*Pold3, Rnf111,* and *Rnf169*[43–50]. The same genes were also upregulated by *Dio2* depletion and at least two of these genes, namely *Mad1l1* and *Pold3,* were direct THs target genes. The same genes are also induced by p53 and confirm a general framework in which loss of p53 increases D2 and the consequent TH activation, thereby lowering the expression of genes involved in DNA repair program (Fig. 8). The role of THs in the DNA damage repair was previously noticed[51], by showing THs induction of DNA DSBs in fibroblasts, leading to premature senescence.

The present work suggests that the effects of THs are quite different in cells with both wild-type copies of p53 or even partial loss of p53. In p53 wild-type cells, the physiological role of THs is directly involved in cell cycle arrest and terminal cell differentiation through the induction of the cell cycle arrest genes *CDKN1A* and *CDKN1B* (encoding for p21-WAF1 and p27-KIP1 proteins, respectively)[4,52–54] or also in increased levels of antioxidant genes, such as *SOD2*[55,56]. In p53-mutated cells, aberrant or persistent THs activation reduces the G1 barrier and provides an escape mode by which damaged cells bypass the physiological checkpoints following DNA damage or inefficient repair and ultimately it facilitates the progress toward malignancy.

Whole-exome and whole-genome sequencing studies of a cohort of SCCs highlighted that, despite the extreme variability in coverage, about 70% of the mutational profile of SCCs harbors recurrent LOF mutations in the *Trp53* gene. In turn, most of these mutations result in altered expression of genes critical for cell-cycle checkpoint control and in the balance between DNA damage repair or the induction of apoptosis, thereby increasing the malignant evolution of cancers[15,16]. The link between p53 LOF and enhanced D2 expression thus reinforces the recent findings that indicate D2 as a metastatic promoter and that hyperthyroidism in patients is associated with advanced tumor staging[57–59].

Thus, in the light of the present work and the previous data in the literature, we can state that, while in poor invasive and most benign tumors, like the BCC, the TH signal is attenuated and the predominant deiodinase is D3[60,61], more invasive tumors with a higher propensity to metastasize the TH and D2 levels are higher[10,59]. The same process can be viewed in the context of the same tumor during the progression, in which the hyperplastic and benign stages are characterized by high D3 expression. In contrast, the evolution toward invasive stages is characterized by high D2 expression[10,59].

Also, how can we explain the action of THs as apoptotic inducers and pro-metastatic agents? At early and mid-stage tumorigenesis, the great majority of tumor cells are experiencing a faulty DDR and undergo apoptosis. At later stages of tumorigenesis, when D2/TH signal is raised by the loss of p53, the few surviving cells are the ones in which DNA repair fails and they accumulate DNA mutations fostering invasive and pro-metastatic features.

Is this true only in transformed cancer cells? We found that in "non-tumorigenic" cells as the HaCaT keratinocytes, p53 downregulates D2 similarly to what observed in tumorigenic cells. Moreover, a physiological condition in which cells accumulate oxidative

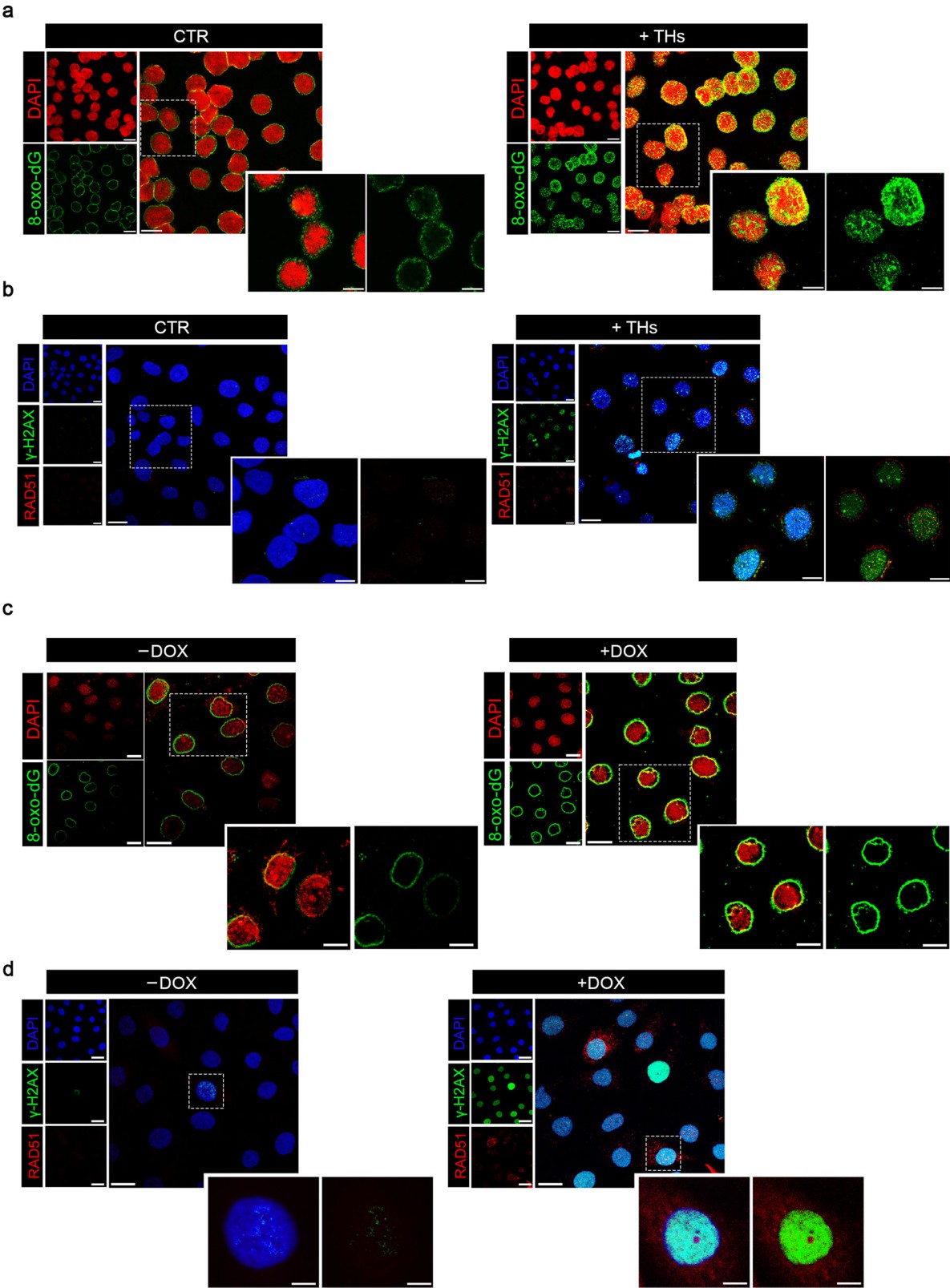

DNA damage occurs during enhanced cellular respiration and mitochondrial production of Reactive Oxygen Species (ROS), leading to both, mitochondrial aberration, and nuclear DNA damage[62,63]. TH action in this respect is peculiar because transcription of some DNA repair genes is inhibited by excess of oxidative local damage, as shown by OGG1 binding to the promoter chromatin of the repair gene we have selected. Accumulation of DSB severely affects the survival of the cells expressing wild-type p53. Loss even of one copy of p53 reduces the selection on the cells, which accumulate overtime when exposed to THs.

The practical implications of these data are important because the evolution and the progression to an invasive phenotype of p53[+/-] epithelial tumors, which may take years to appear clinically, can be slowed down or cured by treating these patients with anti-TH drugs.

**Fig. 6 | THs induce DNA damage and unbalanced DNA damage response.**
**a** Confocal images of 8-oxo-dG IF staining in SCC011 cells treated or not with THs (T3, 30.0 nM + T4, 30.0 nM) for 24 h. Data are presented as overviews (top rows) and higher magnification (bottom rows) (representative of 3 images per sample). Magnification 40X. Scale bars represent 20 μm. **b** Confocal images of γ-H2AX and RAD51 IF staining in SCC011 cells treated or not with THs (T3, 30.0 nM + T4, 30.0 nM) for 24 h. Data are presented as overviews (top rows) and higher magnification (bottom rows) (representative of 3 images per sample). Magnification 40X. Scale bars represent 20 μm. **c** Confocal images of 8-oxo-dG IF staining in SCC13 Tet-ON-D2 cells treated or not with doxycycline (DOX, 2 μg/mL) for 24 h. Data are presented as overviews (top rows) and higher magnification (bottom rows) (representative of 3 images per sample). Magnification 40X. Scale bars represent 20 μm. **d** Confocal images of γ-H2AX and RAD51 IF staining in SCC13 Tet-ON-D2 cells treated or not with doxycycline (DOX, 2 μg/mL) for 24 h. Data are presented as overviews (top rows) and higher magnification (bottom rows) (representative of 3 images per sample). Magnification 40X. Scale bars represent 20 μm.

## Methods

All experiments performed in the current study comply with all relevant ethical regulations.

### Cell cultures, reagents, and treatment

All cell lines used in this paper were correctly authenticated. SCC011 cell line (RRID:CVCL_5986), previously described[64], has been reported having a "damaging" annotation which may lead to either loss-of-function or gain-of-function activities, in our experiments these cells behave as cells with a functional p53, at least for the D2 and p21 regulation, and are then used as a model of cells with a functional p53 protein. SCC011 were cultured in RPMI 1640 Medium (HiMedia Leading BioSciences Company, Mumbai, Maharashtra, India, cod. AL028) supplemented with 10% Fetal Bovine Serum (HiMedia Leading BioSciences Company, Mumbai, Maharashtra, India, cod. RM10432), 1% L-Glutamine (Gibco, Thermo Fisher Scientific, Waltham, MA, USA, cod. 25030024) and 1% Penicillin/Streptomycin (Gibco, Thermo Fisher Scientific, Waltham, MA, USA, cod. 15070063). p53 mutant SCC cell line (SCC13, RRID:CVCL_4029) and p53 mutant D2 TET-ON SCC cell line[12], derived from a skin SCC[65], were cultured in Keratinocyte-SFM (KSFM 1X, Gibco, Thermo Fisher Scientific, Waltham, MA, USA, cod. 17005059) medium [+] L-Glu (Gibco, Thermo Fisher Scientific, Waltham, MA, USA) with Bovine Pituitary Extract (BPE, 30.0 μg/mL, Gibco, Thermo Fisher Scientific, Waltham, MA, USA, cod. 11543530) and human recombinant Epidermal Growth Factor (EGF, 0.24 ng/mL, Gibco, Thermo Fisher Scientific, Waltham, MA, USA, cod. 11543530). HaCaT cells (RRID:CVCL_0038) were cultured in Dulbecco's modified Eagle medium (DMEM, HiMedia Leading Bio Sciences Company, Mubai, India, cod. AL007) supplemented with 10% Fetal Bovine Serum, 1% L-Glutamine (Gibco, Thermo Fisher Scientific, Waltham, MA, USA, cod. 25030024) and 1% Penicillin/Streptomycin. All cell lines were mycoplasma free and were cultured at 37 °C in a humidified atmosphere with 5% CO₂. In all the experiments in which THs was applied to cells, we used a combination of T3 (30.0 nM/24 h, Sigma-Aldrich, St. Louis, Missouri, USA, cod. T6397) and T4 (30.0 nM/24 h, Sigma-Aldrich St. Louis, Missouri, USA, cod. T2501), indicated throughout the text as THs, thus resembling physiological exposure of cells to both the active hormone (T3) and its pro-hormone (T4). In experiments in which THs were removed from the serum, THs-deprivation was achieved by FBS Charcoal absorption. For the studies of DNA damage mechanisms, we used etoposide (50.0 μM/30 min, Sigma-Aldrich, St. Louis, MO, USA, cod. E1383) and KuDOS (KU-55933, 100.0 μM/1 h, Sigma-Aldrich, St. Louis, MO, USA, cod. SML1109). UV-A/UV-B radiation was performed by exposing cells to 60 mJ/cm² UV-A/UV-B light, using six Philips TL12/60W fluorescent lamps (Philips, Eindhoven, The Netherlands). UV-C radiation was performed by exposing cells to 100 μJ/cm² UV-C light, using UV Stratalinker 2400 (Agilent Genomics/Stratagene Stratalinker 1800 UV Crosslinker, Stock #53267-1).

### Plasmids and transfections

The wild-type p53-FLAG plasmid and the plasmids carrying p53 DNA-contact mutations (R248W, E258K, P278F, and E286K) were kindly provided by Prof. Caterina Missero (University of Naples, "Federico II"). The negative control plasmid, namely CMV-FLAG, as well as all the parental reporter plasmids, namely Dio2-LUC-WT, TRE3-TK-LUC, and CMV-Renilla, were already available in the laboratory. Non-targeting control and human p53-specific stealth (p53-stealth-1 sequence: CGAUAUUGAACAAUGGUUCACUGAA; p53-stealth-2 sequence: UUCAGUGAACCAUUGUUCAAUAUCG) for RNA interference (RNAi) experiments were purchased from Sigma-Aldrich (Sigma-Aldrich, St. Louis, MO, USA). All transient transfections were performed using Lipofectamine-3000 (Invitrogen™, Carlsbad, CA, USA, cod. L3000015), according to the manufacturer's instructions.

### Luciferase (Luc) expression assay

The reporter plasmids (Dio2-LUC-WT, Dio2-LUC-Δ1, Dio2-LUC-Δ2, Dio2-LUC-Δ3, Dio2-LUC-mut, TRE3-TK-LUC) and CMV-Renilla were co-transfected into SCC cells. LUC activities were measured 48 h after transfection with the Dual-Luciferase Reporter Assay System (Promega, Madison, Wisconsin, USA, cod. E1910). Luminescence was measured with a Lumat LB Single Tube Luminometer (Berthold). Differences in transfection efficiency were corrected relative to the level of Renilla Luciferase. Each construct was studied in triplicate in at least three separate transfection experiments.

### DIO2 targeted mutagenesis

Targeted mutagenesis of DIO2 gene in SCC011 cells was achieved by using the CRISPR/Cas9 system from Santa Cruz Biotechnology (DIO2 CRISPR/Cas9 KO Plasmid (h), cod. sc-402262). Control SCC cells were stably transfected with the CRISPR/Cas9 control plasmid (Control CRISPR/Cas9 Plasmid, cod. sc-418922). 48 h after transfection with CRISPR/Cas9 plasmids, the cells were sorted using Fluorescence Activated Cell Sorting (FACS) for green fluorescent protein expression. Single clones were analyzed by PCR to identify alterations in coding regions, and DIO2 exon 1 was sequenced to identify the inserted mutations. All the experiments in D2KO cells were repeated in three different clones to avoid off-target effects.

### Apoptosis detection through the Annexin V-FITC staining

Apoptosis was induced by using chemical DNA damaging agent etoposide and physical damaging agent UV-C radiation. Apoptosis analyses were performed by using the Annexin V-FITC Apoptosis Detection kit (Sigma-Aldrich, St. Louis, MO, USA, cod. APOAF) according to the manufacturer's instructions. Stained cells were analyzed on a BD FACS Canto flow cytometer using BD FACS Diva software (Duke University Flow Cytometry Shared Facility), and data were analyzed using FlowJo (see Supplementary Figs. 17, 18).

### In silico promoter analysis for searching transition factor binding sites

Consensus p53 binding sites (AC M00272, ID V\$P53_02) with matrix similarity scores of 0.75 or greater (maximum 1.00) in the upstream region of the murine Dio2 gene promoter were identified using TFBIND (https://tfbind.hgc.jp). Position analyses of all identified consensus p53 binding sites were reported in Supplementary Fig. 1. The analysis was also performed by using LASAGNA-Search 2.0: Searching for Transcription Factor Binding Sites (TFBSs) (https://biogrid-lasagna.engr.uconn.edu/lasagna_search/index.php), in which are enclosed TRANSFAC, JASPAR, UniPROBE, and ORegAnno matrices. Similarly, we searched consensus thyroid hormone receptor binding sites (TREs, AC

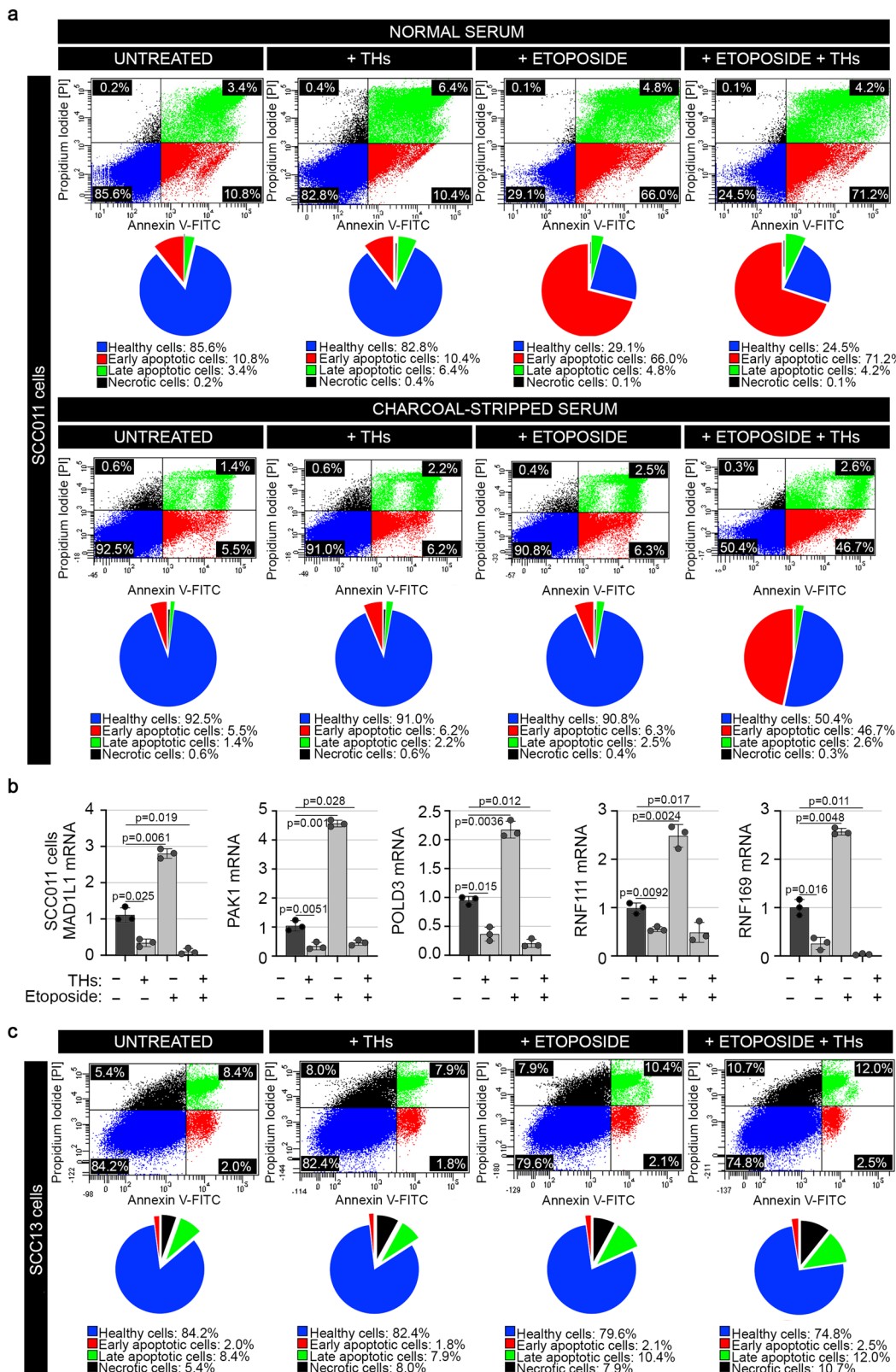

M00239, ID V$T3R_01) in the upstream promoter region of *MAD1L1*, *PAK1*, *POLD3*, *RNF111,* and *RNF169* genes (Supplementary Figs. 12–14).

### RNA extraction and real-time PCR

Messenger RNAs were extracted with TRIzol reagent (Life Technologies Ltd, Carlsbad, CA, USA, cod. 15596018). Complementary DNAs (cDNAs) were prepared with SuperScript™ VILO™ MasterMix (Life Technologies Ltd, Carlsbad, CA, USA, cod. 11755-050) as indicated by the manufacturer. The cDNAs were amplified by PCR in a CFX Connect Real-Time PCR Detection System (BioRad, Hercules, California, USA, cod. 1855201) with the fluorescent double-stranded DNA-binding dye SYBR Green (BioRad, Hercules, California, USA, cod. 1708882). Specific primers for each gene were designed to work under the same cycling conditions (95 °C for 10 min followed by 40 cycles at 95 °C for 15 sec

**Fig. 7 | THs alter cell viability and induce cell apoptosis. a** Flow cytometric analysis for monitoring the effect of THs-induced apoptosis in SCC011 cells, treated or not with etoposide in Normal Serum and Charcoal-Stripped Serum. Pie charts represent percentages of Viable (V), Early (E), and Late (L) apoptotic, and Necrotic (N) cells measured by Annexin V-FITC/PI co-staining. Graphs represent an average of 3 separate experiments. **b** mRNA expression of *MAD1L1*, *PAK1*, *POLD3*, *RNF111*, and *RNF169* was measured by Real-Time PCR in SCC011, treated or not with THs (T3, 30.0 nM + T4, 30.0 nM) for 24 h, alone and in combination with etoposide

(50.0 μM/30 min) (*n* = 3 independent experiments). All the results are shown as means ± SD from at least 3 separate experiments. *p*-values were determined by two-tailed Student's t-test. \**p* < 0.05, \*\**p* < 0.01, \*\*\**p* < 0.001. **c** Flow cytometric analysis for monitoring the effect of THs-induced apoptosis in SCC13 cells, treated or not with etoposide. Pie charts represent percentages of Viable (V), Early (E), and Late (L) apoptotic, and Necrotic (N) cells measured by Annexin V-FITC/PI co-staining. Graphs represent an average of 3 separate experiments. Source data are provided as a Source data file.

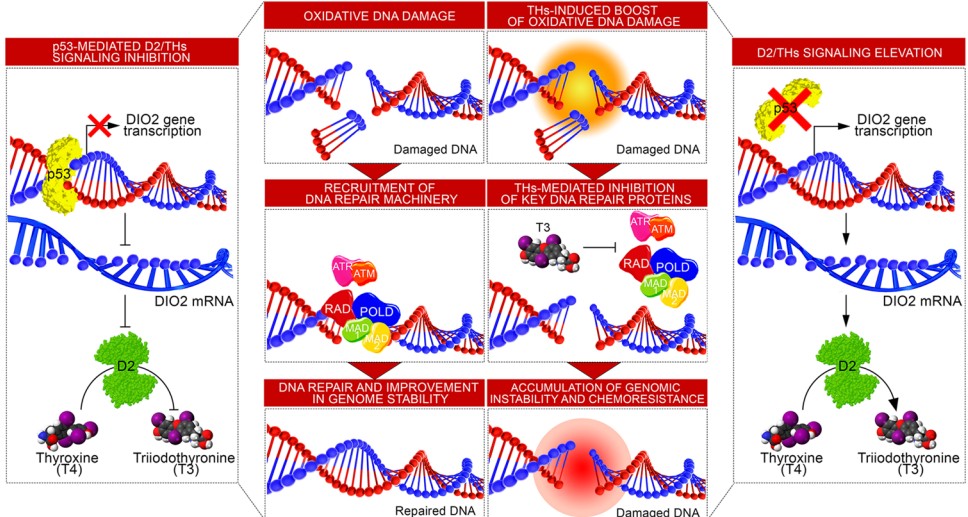

**Fig. 8 | Loss of p53 leads to D2 re-expression and enhanced DNA damage.** Graphical illustration of the p53-mediated D2 regulation and the transcriptional program leading to modulation of genes involved in the DNA Damage Response (DDR).

and 60 °C for 1 min), thereby generating products of comparable sizes (about 200–300 bp for each amplification). Primer combinations were positioned whenever possible to span an exon-exon junction and the RNAs were digested with DNAse to avoid genomic DNA interference. Primer sequences are indicated in Supplementary Data 2. For each reaction, standard curves for reference genes were constructed based on six four-fold serial dilutions of cDNA. All samples were run in triplicate. The template concentration was calculated from the cycle number when the amount of PCR product passed a threshold established in the exponential phase of the PCR. The relative amounts of gene expression were calculated with Cyclophilin-A (CyA) expression as an internal standard (calibrator). The results, expressed as N-fold differences in target gene expression, were determined as follows:
N*target = $2^{(\Delta Ct\ sample-\Delta Ct\ calibrator)}$.

**Protein extraction from cells and skin and western blot analysis**
Dorsal skin was removed from mice and immediately snap-frozen in liquid nitrogen. 800 μL of Lysis Buffer (0.125 M Tris pH 8.6; 3% SDS, protease inhibitors including PMSF 1.0 mM and phosphatase inhibitors) were added to all dorsal skin samples, which were then homogenized with TissueLyser II (QIAGEN, Hilden, Germany, cod. 85300). Total tissues protein or cell protein was separated by 8-10 or 15% SDS-PAGE followed by western blot. The membrane was then blocked with 5% non-fat dry milk (BioRad, Hercules, California, USA, cod. 1706404) in PBS-0.2% Tween. The following primary antibodies and dilutions were used: anti-ZEB1 (abcam, cod. ab-155249), anti-E-Cadherin (BD Biosciences, cod. 610181), anti-Vimentin (abcam, cod. ab-92547), anti-Slug (Cell Signaling, cod. 9585S), anti-Snail (Cell Signaling, cod. 3895S), anti-ATM (abcam, cod. ab199726), anti-ATM phospho-S1981 (abcam, cod. 81292), anti-p53 (Cell Signaling, cod. 2524S), anti-p53 phospho-Ser15 (Cell Signaling, cod. 9284S), anti-p21 (Santa Cruz Biotechnology, cod. sc-397), anti-BCL2 (Cell Signaling, cod. 2872S), anti-BAX (Cell

Signaling, cod. 2772S), anti-Cyclin-D1 (Santa Cruz Biotechnology, cod. sc-246), anti-PCNA (Santa Cruz Biotechnology, cod. sc-7907), anti-p38 MAPK (Cell Signaling, cod. #9212), anti-phospho-p38 MAPK (Thr180/Tyr182) (Cell Signaling, cod. #9211S), anti-ERK1/2 (K-23) (Santa Cruz Biotechnology, cod. sc-94), anti-phospho-p44/42 MAPK (ERK1/2) (Thr202/Tyr204) (Cell Signaling, cod. #4370), anti-GAPDH (Elabscience, cod. E-AB-20059) and anti-Tubulin (Santa Cruz Biotechnology, cod. sc-5546) antibodies O/N at 4 °C. The following secondary antibodies were also used: anti-mouse IgG-HRP (BioRad, Hercules, California, USA, cod. 1706516) and anti-rabbit IgG-HRP (BioRad, Hercules, California, USA, cod. 1706515) and detected by chemiluminescence using an ECL kit (Millipore, Burlington, Massachusetts, USA, Cat. No. WBKLS0500). Anti-Tubulin or anti-GAPDH specific antibodies were used as a loading control. All Western Blots were run in triplicate, and bands were quantified with ImageJ software (NIH Image, Bethesda, Maryland). For antibodies dilutions see Supplementary Data 3.

**Chromatin immuno-precipitation (ChIP) assay**
SCC cells were (i) transfected with p53-FLAG plasmid or a negative control plasmid, i.e., CMV-FLAG, for 48 h (conditions for experiments showed in Fig. 1k) and (ii) treated with THs (T3, 30.0 nM/24 h + T4, 30.0 nM/24 h, conditions for experiments showed in Fig. 5e). Approximately 2 × 10⁶ cells were fixed for 10 min at 37 °C by adding 1% formaldehyde (Sigma-Aldrich, St. Louis, MO, USA, cod. 1004968350) in growth medium. The reaction was quenched by the addition of glycine to a final concentration of 0.125 M. Fixed cells were harvested and the pellet was resuspended in 1.0 mL of Lysis Buffer containing protease inhibitors [200.0 mM Phenyl-Methyl-Sulfony Fluoride (PMSF, Sigma-Aldrich, St. Louis, MO, USA, cod. P7626), 1.0 μg/mL Aprotinin (Sigma-Aldrich, St. Louis, MO, USA, cod. A6279)]. The lysates were sonicated to obtain DNA fragments of 200–1000 bp. Sonicated

samples were centrifuged, and the soluble chromatin was diluted 10-fold in dilution buffer and used directly for ChIP assays. An aliquot (1/10) of sheared chromatin was further treated with Proteinase K (Thermo Fisher Scientific, Waltham, MA, USA, cod. 25530), extracted with phenol/chloroform and precipitated to determine DNA concentration and shearing efficiency ("input DNA"). Briefly, the sheared chromatin was pre-cleared for 2 h with 1.0 µg of non-immune IgG (Calbiochem, from Sigma-Aldrich, St. Louis, MO, USA) and 30.0 µL of Protein-G Plus/Protein-A Agarose suspension (GE Healthcare, from Sigma-Aldrich, St. Louis, MO, USA, cod. GE17-0780-01) saturated with Salmon Sperm (1.0 mg/mL, Sigma-Aldrich, St. Louis, MO, USA). Pre-cleared chromatin was divided in aliquots and incubated at 4 °C for 16 h with 2.5 µg of antibody (anti-FLAG, Sigma-Aldrich, St. Louis, MO, USA, cod. F3165; anti-Thyroid hormone receptor antibody (C3)-Chip Grade, Abcam, cod. ab-2743; anti-Thyroid hormone receptor beta antibody Chip Grade, Abcam, cod. ab-5622; anti-OGG1/2, Sigma-Aldrich, St. Louis, MO, USA, sc-376935—see Supplementary Data 3). After five rounds of washing, bound DNA-protein complexes were eluted by incubation with 1% Sodium Dodecyl Sulfate-0.1 M NaHCO$_3$ Elution Buffer. Formaldehyde cross-links were reversed by incubation in 200.0 mM NaCl at 65 °C. Samples were extracted twice with phenol-chloroform and precipitated with ethanol. DNA fragments were used for Real-Time PCRs. For primer sequences see Supplementary Data 2.

### Immunofluorescence

For immunofluorescence staining, SCC011 cells were grown on cover slips until the reaching of 80% of confluence, fixed with 100% cold methanol on ice, and permeabilized in 0.1% Triton X-100 for 15 min. After permeabilization, cells were blocked with 0.3% BSA/PBS and washed in PBS. Next, cells were incubated with primary γ-H2AX (anti-phospho-Histone H2A.X (Ser 139) clone JBW301, Cat. 05-636, Millipore, Burlington, Massachusetts, USA) and RAD51 (Rad51 (H-92), cod. sc-8349, Santa Cruz Biotechnology) antibodies (for antibodies dilutions see Supplementary Data 3), O/N at 4 °C. Secondary antibody incubation was carried out at room temperature for 1 h, followed by washing in 0.2% Tween/PBS. Images were acquired with a ZEISS LSM 900 Airyscan 2 confocal microscope. For 8-oxo-dG assay, the immunofluorescence staining was performed according to the manufacturer's instructions, by using the anti-8-oxo-dG antibody (TREVINGEN, cod. 4354-MC-050). The measure of the co-localization of 8-oxo-dG/DAPI signals was performed by counting the number of nuclei in a field. Briefly, after counting the number of nuclei in each field, we determine the number of foci, normalizing for the number of nuclei in each field. The foci in each region were defined by the nuclear staining overlap.

### Animals, histology, and immunostaining

p53KO$^{+/-}$;D2KO (K14-Cre$^{ERT}$;p53KO$^{+/-}$;D2$^{fl/fl}$) mice were obtained by crossing the keratinocyte-specific conditional K14-Cre$^{ERT}$ mouse[27] with D2$^{fl/fl}$ mice[26]. Dio2 genetic depletion was induced by treatment with 10.0 mg of Tamoxifen (Tamoxifen free-base, Sigma-Aldrich, St. Louis, MO, USA, cod. T5648) at two-month-old mice, as indicated in Fig. 3a. Skin lesions were harvested at 20 weeks after Tamoxifen administration and DMBA treatment. Hyperthyroid mice were obtained by treating 12-weeks-old K14-Cre$^{ERT}$;p53WT;D2WT male mice with T3 (1.0 mg/mL) and T4 (4.0 mg/mL) in drinking water for 3 weeks. For immunofluorescence and histology, dorsal skin from p53KO$^{+/-}$;D2WT, p53KO$^{+/-}$;D2KO, and control mice (p53WT;D2WT) was embedded in paraffin, cut into 7.0 µm sections, and H&E-stained. Slides were baked at 37 °C, deparaffinized by xylenes, dehydrated with ethanol, rehydrated in PBS, and permeabilized by placing them in 0.2% Triton X-100 in PBS. Antigens were retrieved by incubation in 0.1 M citrate buffer (pH 6.0) or 0.5 M Tris buffer (pH 8.0) at 95 °C for 5 min. Sections were blocked in 1% BSA/0.02% Tween/PBS for 1 h at room temperature. Primary antibodies were incubated O/N at 4 °C in blocking buffer and washed in 0.2% Tween/PBS (for antibodies dilutions see Supplementary Data 3). Secondary antibodies were incubated at room temperature for 1 h and washed in 0.2% Tween/PBS (for antibodies dilutions see Supplementary Data 3). Images were acquired with a Leica DMi8 microscope and the Leica Application Suite LAS X Imaging Software.

### Cutaneous chemical carcinogenesis and animal study approval

All animal studies were conducted in accordance with the guidelines of the Ministero della Salute and were approved by the Institutional Animal Care and Use Committee (IACUC, nos. 167/2015-PR and 354/2019-PR). The dorsal skin of 2-months-old male mice was treated with a dose of the carcinogen 7,12-dimethylbenz[a]anthracene (DMBA, Sigma-Aldrich, St. Louis, MO, USA, cod. D3254, 100.0 µL, 1.0 mg/mL) resuspended in propanone twice a week for 8 weeks. Experiments were performed using 12 CTR mice (p53WT;D2WT), 12 p53KO$^{+/-}$;D2WT mice and 12 p53KO$^{+/-}$;D2KO mice. Tumor growth was examined by measuring the greatest longitudinal diameter (length) and the greatest transverse diameter (width) with digital calipers, and tumor volume (mm$^3$) was calculated using the formula $\pi/6 \times$ larger diameter $\times$ (smaller diameter)$^2$. Tumors were measured once every 2 weeks. All tumor measurements within single cohorts were performed by the same researcher. Mice were euthanized by CO$_2$ inhalation when tumor volumes met humane endpoints described in the IACUC protocols or upon severe health deterioration. The maximum tumor diameter permitted under the relevant animal protocols is 20 mm, and this limit was not exceeded in any experiment.

### Isolation and analysis of CTCs from mice blood samples

Mice blood samples (500 µL) were collected in RNA Protect Animal Blood Tube and stored at room temperature up to 30 min. RNA was purified using the RNeasy Protect Animal Blood Kit (QIAGEN, Hilden, Germany, Cat. No. 73224), and Krt8, Krt19, and c-Met expression levels were analyzed by Real-Time PCR. For primer sequences see Supplementary Data 2.

### RNA-seq following LCM, Gene Ontology analysis, and pathway enrichment analysis of DEGs

Total RNA was extracted from the epithelial compartment of SCC tumors isolated from dorsal skin of p53KO$^{+/-}$;D2KO versus p53KO$^{+/-}$;D2WT mice using the RNeasy kit (QIAGEN, Hilden, Germany), dissolved in RNase-free water and quantified using the Qubit 2.0 Fluorimetric Assay (Thermo Fisher Scientific). RNA samples were sequenced by Next Generation Sequencing - TIGEM|Bioinformatics Core. Libraries were prepared from 125.0 ng of total RNA using the 3'DGE mRNA-seq research grade sequencing service (Next Generation Diagnostics), which included library preparation, quality assessment, and sequencing on a NovaSeq 6000 sequencing system using a single-end, 100 cycle strategy (Illumina Inc.). The raw data were analyzed by Next Generation Diagnostics s.r.l. proprietary 3'DGE mRNA-seq pipeline (v2.0), which involves a cleaning step by quality filtering and trimming, alignment to the reference genome, and counting by gene. The raw expression data were normalized, analyzed, and visualized by Rosalind HyperScale architecture (OnRamp BioInformatics, Inc.). The Gene Set Enrichment Analysis (GSEA) software, a joint project of UC San Diego and Broad Institute (http://www.gsea-msigdb.org/gsea/index.jsp), was used to explore the Molecular Signatures Database (MSigDB), to investigate the gene sets in the online biological network repository NDEx. FDR q-value less than 0.05 was considered to indicate a statistically significant difference.

### Statistics

Statistical analyses were performed with Microsoft Excel (version 16.61). For the experiments in which we compared two different conditions, Student's two-tailed t test or Mann-Whitney test were applied. For the experiments in which multiple conditions were compared, we

**Article**

used the one-way ANOVA test. All the data were obtained by performing at least 3 independent experiments and are expressed as means ± Standard Deviation (SD) of 3 independent experiments. The reproducibility, sample sizes and, where appropriate, statistical analyses are described in the figure legends. Relative mRNA levels (in which the control sample was arbitrarily set as 1) are reported as results of Real-Time PCR, in which the expression of Cyclophilin-A (CyA) served as housekeeping gene. A *p*-value <0.05 was considered statistically significant (Confidence level: 95%).

### Reporting summary

Further information on research design is available in the Nature Portfolio Reporting Summary linked to this article.

## Data availability

Data supporting the findings of this work are available within the paper and Supplementary Information files. All data from this study have been submitted to the NCBI Sequence Read Archive (SRA). The sequencing data generated for this study have been deposited in NCBI Sequencing Read Archive (SRA) database under the BioProject accession number PRNA891519 (https://www.ncbi.nlm.nih.gov/bioproject/PRJNA891519). The publicly available gene expression residual dataset used in this study is available in GEO Dataset GSE42677 (https://www.ncbi.nlm.nih.gov/geo/query/acc.cgi?acc=GSE42677). The remaining data are available within the Article, Supplementary Information or Source data file provided with this paper. Source data are provided with this paper.

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

## Acknowledgements

This work was supported by grants from AIRC to M.D. (IG 13065) and by the grant PRIN from MIUR (2017WNKSLR) to M.D. A.N. was supported by an AIRC fellowship for Italy grant (project code 26823). The authors have declared that no conflict of interest exists. E.A. was supported by European Union's Horizon 2020 Research and Innovation Program under the Marie Skłodowska-Curie grant agreement (No 671881 INTEGRATA).

## Author contributions

A.N., C.M., A.P., A.T., E.D.C., S.S., A.G.C., M.M., S.T., and E.A. performed in vitro and in vivo experiments. G.C. performed experiments of UV-radiation; A.N. generated the plasmids and prepared figures; A.N., C.M., and E.D.C. performed the immunofluorescence analysis; M.R. performed the FACS analysis studies; M.S. performed confocal analysis; D.A. performed bioinformatic analysis, analyzed the results and provided scientific interpretations; A.N. and M.D. wrote the paper; G.F., D.S., and V.E.A. supervised the experiments, analyzed the results and provided scientific interpretations; M.D. designed the overall study, supervised the experiments and analyzed the results. All authors discussed the results and provided input on the manuscript.

## Competing interests

The authors declare no competing interests.
