## [Peer Review File · Nature Communications]

Reviewers' Comments:

Reviewer #1:

Remarks to the Author:

In this research work, Nappi et al, demonstrated that D2 is a novel p53-repressed target-gene in SCC. This conclusion is solidly based on several complementary methods including in vitro and in vivo data, which are a prerequisite for establishing/identifying genes as new p53-targets. The research work is comprehensive and systematic and the text is very well written, in a clear and organized manner. There are several major issues that need to be addressed/explained/discussed before considering publication at Nature Communications:

Major comments:

1. I agree that you have demonstrated that p53 suppresses D2 and thereby in p53 mutated SCC cells, it results in high (non-inhibited) D2 mRNA/protein levels and that this contributes to tumor progression. However, if this was the canonical pathway in all aggressive tumors, the majority of which are p53 mutated, we would expect high expression of D2 in all tumor types, as well as a correlation with worse survival. Analyses of the TCGA data, protein atlas and survival plots for D2, for an array of malignancies, clearly indicates that high D2 is not a general phenomenon. For example in several tumors D2 is hardly expressed and in colon/endometrium high D2 is a "good prognostic factor". So D2 expression is highly heterogeneous and dynamic between tumors. What I suggest is to "de-generalize" the discovery. It should be clear in the text (abstract, introduction and discussion) that this association is established in SCC, but not in all tumors.

2. Based on your previous works, it seems that D2 acts opposite in BCC compared to SCC. In addition you demonstrated that the balance between D2 and D3 is different in both carcinomas. This should be discussed, as both models are presumably mutated in p53, but the outcome seems to be different.

3. You did not show D1 and D3 levels in the cell models/mice you used. This raises several questions: Did you search for p53-RE in D1 and D3? Can there be a compensation mechanism by D1 when D2 is inhibited by WT p53? What happens to D3 in your samples- as you suggest that there is increased T3-bioavailability when p53 is mutated, is D3 repressed or not expressed at all in these SCC-manipulated cells?.

4. One major issue is the fact that you refer to the SCC011 cells as "p53 WT". You cited ref #18 when referring to SCC011 as being p53 WT, but I could not find these specific cells mentioned in that specific reference. You further cite from ref. 59 "p53 wild-type SCC cell line (SCC011), previously described". In this ref. 59 it is written that the cells, which are also called "JHU0011", were given by Dr. J. Rocco. I searched and noticed that Rocco got the cells from David Sidransky, JH University (p63 mediates survival in squamous cell carcinoma by suppression of p73-dependent apoptosis. James W. Rocco et al, CANCER CELL 9, 45-56, 2006). These JHU0011/SCC011 cells are described at expasy: https://web.expasy.org/cellosaurus/CVCL_5986), as well as in earlier work by Sidranski. Anyhow, in all published works these cells are reported as p53 truncated mutants, resulting from a splice donor mutation following exon 6 and are therefore p53 null.

I found another line which is also termed SCC011" and is from the same origin (Laryngeal SCC): https://web.expasy.org/cellosaurus/CVCL_7805 University of Turku-Squamous Cell Carcinoma-11, . These SCC011 cells are also p53 mutated, but with a deletion in AA187-196 in the DBD. This information is also nicely presented in figure 2 here: Genotyping and Characterization of HPV Status, Hypoxia, and Radiosensitivity in 22 Head and Neck Cancer Cell Lines, Göttgens et al., Cancers 2021, 13, 1069; <https://doi.org/10.3390/cancers13051069>).

If the cells you used are from a different origin than the ones I found in the reference you cited, than disregard my comments and provide the exact cell source, otherwise, this means that your results are based on mutated p53. The good news is that you used p53-transfected SCC011 cells in many experiments in figure 1, so most results remain validated as you rescued the WT p53 and it inhibited D2. Anyhow, the exact cell line and its p53 status should be validated and if indeed mutated, corrected throughout the manuscript.

5. Figure 6:

>You use 30nM T3 concentrations which are considered in vitro supra-physiological by any

measure (e.g Endocrinology, November 2015, 156(11):4325– 4335, doi: 10.1210/en.2015-1221). For near physiologic concentrations most groups use ~1nM T3. Unless you have indications showing that in your cell system under these conditions T3 levels are within the normal range, it should be clear that these are not physiological levels in the text.

>By adding TH to the cell medium, you can actually activate the non-genomic avb3 integrin axis. Unless you can validate that the cells are avb3 negative, it cannot be fully ruled out that the results you received by adding exogenous T3 are non-genomic-mediated or D2-mediated. This possibility should be explained in the text and discussed.

>The % of changes in the Annexin-PI assay are extremely low and only representative histograms are shown.

>The only panel that makes sense is panel d, but again, there is no guaranty that the results are mediated directly via the p53-D2 axis.

Therefore, the way I see it, the work presented in figure 6 weakens your very strong evidence presented in figures 1-5. I would omit it from this current work, unless you can substantiate the results based on the integrin expression and repeat the apoptosis assays to achieve more meaningful results.

Minor comments:

1. Manuscript title: I would mention D2 in the title, as it is the essence of this research work and it is important to convey the message that it is a novel p53-target gene. In addition, the oxidation part in the title is based on figure 6, which, as I mentioned in the comment above, demonstrates very limited results, which, in contrast to the remaining work, are not established as direct via the D2-p53 axis.

2. Figure 2:

>In Fig. 2e, the mutant p53 in SCC013 (E286G) is in the same AA site as one of the mutated p53 constructs you used in Fig. 2b-d (E286K). This might be a good addition for explaining the selection of these specific cells which carry an endogenous mutation at a similar DBD site. This also gives clinical relevance to your mutated construct and gives validation that the end of the DBD region is critical for p53 binding to the D2 promoter.

>In Fig. 2g, although it is written that no effect was observed on D2 following UV irradiation in the p53-mutated cells, it is quite clear that there is a trend of elevation up to 8 hr. Maybe it did not reach statistical significance, but the trend is there and should be discussed or at least mentioned.

3. Figures 3 and 4:

> You generated heterozygous Trp53 gene deletion in the mice, but homozygous for D2. Does this mean that loss of one allele is sufficient to lose p53 activity towards D2 promoter (some sort of haploinsufficiency)? Please explain and if relevant discuss in the text.

> Figure 3K and 3M. You show changes in several proteins by WB/IHC, respectively, but you don't show D2 itself. I understand that you provided the RNA level in the supplementary figure, but adding the protein level would add a lot to the figure validity (as you previously showed in Miro et al., 2019).

> Similarly in Figure 4G- D2 IHC will be a good addition.

4. Supplemental figures:

> Supp Figure 3A, right panel- what is D2-REC? it is not mentioned anywhere in the text and should be explained in the figure legend.

>Supp 9b- The % of the HR reduction are minute. What is the meaning of such small changes?

Reviewer #2:

Remarks to the Author:

This manuscript by Nappi and colleagues describes a mechanism for regulating levels of thyroid hormone activating enzyme type 2 deiodinase by p53. The authors cite a significant body of previous work indicating the T3 levels and hyperthyroidism are associated with increased incidence and more aggressive biology in some cancers.

These studies lead to the hypothesis that increased D2/TH signaling in cancer cells leads to tumor

progression and aggressive biology. Therefore what mechanism is responsible for increased D2/TH signaling in cancers.

To evaluate this hypothesis the investigators conducted an in silico analysis and identified potential binding sites for p53 in the Dio2 promoter region which was supported by an inverse correlation between TP53 and ATM and DIO2 expression. The range of assays used including a combination of biochemical assays on cell lines and genetic models confirm that p53 binds to a predicted P1 sequence in the promoter to suppress D2 mRNA production.

Although much of the data appear reasonable there is a major experimental design flaw that negatively impacts the relevance of these data to understanding UV-induced skin cancer. The manuscript is well written.

Major Issue: The UV irradiation is performed using a Stratlinker. Stratlinkers emit predominantly UVC which is not a relevant cause of UV-induced skin cancer because it is absorbed by the atmosphere and does not reach the earth's surface. These experiments should be done using a mixture of UVB and UVA emitting lamps (there are many available). UVC filters should be in place to eliminate any UVC that may be emitted.

Other issues mentioned below should be addressed.

1. The authors tend to overstate the significance of Dio2 in p53-regulated tumorigenesis. Though the data are intriguing, they show that Dio2 is a downstream effector of p53 and likely a key regulator of metabolism in neoplastic cells.

Downregulation of Dio2 is dependent on p53, the authors infer that DNA damage is regulated by Dio2 levels. This is speculative because they have not shown that this is independent of p53 loss of function.

Sentences addressing this in lines 354-364 should be revised, note that DSBs typically do not occur during UV-treatment of keratinocytes. NER and SS DNA breaks occur. Not clear that TH induced DSBs are relevant to UV induced skin cancer biology.

2. The authors focus on potential increased metastasis as a result of disrupted p53/D2 signaling which is important. However, they show this pathway is disrupted in SCCIS and since SCCIS have been known to have classic oncogenic p53 mutations in approx. 60% of lesions (PMID 32649944) the biological impact of disrupted p53/D2 signaling would be seen in a majority of these lesions.

This should be discussed as SCCIS are much more common than cSCCs .

3. The 8oxodG staining technique should be clarified, were tissue sections protected from atmospheric oxygen which promotes 8oxodG formation? Also in the control panels in Fig 6a, the 8oxodG appears not to overlap with the DAPI stain. Is that all mitochondrial DNA staining? If the data is to be shown there should be some type of quantitative analysis.

Reviewer #3:

Remarks to the Author:

This study shows that the tumor suppressor p53 downregulates expression of type 2 Deiodinase (D2) which catalyzes conversion of thyroid hormone T4 to T3 and thereby increases availability of active thyroid hormone. The authors demonstrate that p53 inactivation leads to elevated D2 expression and increased fitness of tumor cells via regulation of genes involved in DNA damage and repair and redox signaling. This suggests that D2 or thyroid hormone could be relevant targets for treatment of TP53 mutant tumors. The data presented in this study provide a novel understanding of the impact of TP53 mutations in tumor progression and clearly have very interesting implications for novel cancer therapy.

Specific comments:

1. Overall, the data are solid and support the author's conclusions. However, one problem is that p53-mediated suppression of transcription is still poorly understood at the molecular level and highly controversial. While it is well established that p53 binding to DNA motifs is essential for p53-mediated transcriptional transactivation of classical target genes such as p21, Bax and Puma, the molecular basis for p53-mediated suppression of transcription has not been completely elucidated. The authors show that D2 has two p53 binding motifs in its promoter and that P1 is critical for suppression. Yet it is unclear why p53 binding to the D2 promoter leads to suppression of transcription whereas p53 binding to for instance the p21 promoter is known to activate transcription. The authors should discuss this point.

2. In agreement with the general concept of p53-dependent transactivation of transcription, mRNA levels for p53 target genes p21 and Bax are induced upon UV treatment in TP53 wild type SCC cells (Suppl Fig. 2a). However, it is surprising that the canonical p53 target gene MDM2 is strongly suppressed. Could the authors provide a plausible explanation?

3. The authors have examined D2 expression in the presence and absence of functional p53 in UV-treated SCC011 (wt p53) and SCC13 (mut p53) cells in Fig. 2. However, the exact role of D2 and TH in the context of a p53-mediated cellular stress response in normal cells has not been examined. Does activation of wt p53 by DNA damage in normal cells affect D2 and DEGs in a similar way as shown for the SCC cells? This could be tested in for example human diploid fibroblasts or primary keratinocytes.

4. The defective DNA repair and enhanced DNA damage in cells with elevated THs is intriguing but no data on p53 and D2 are shown (Fig. 6). Is p53 induced by etoposide in the SCC011 cells? Is D2 downregulated? SCC13 cells with mutant p53 should be tested in parallel for comparison.

5. According to Fig. 2h, p53 levels are high even in the untreated SCC011 cells, there is no significant further accumulation of p53 upon UV irradiation, although p53 Ser-15 phosphorylation is increased at 4 and 8 hours. Could the authors comment on this?

6. Suppl Fig. 5e needs some clarification. The figure shows reduced expression levels of the five top DEGs in tumors from mice with THs in the drinking water. Does the figure show ratio between expression levels in the D2KO and D2WT tumors with and without THs?

7. The five identified DEGs all have multiple TREs (Suppl Figs. 6-8). Yet according to the CHIP data in Fig. 5e TRs bind physically only to MAD1L1 and POLD3. What is the reason for the differential binding despite the presence of TREs? How are PAK1, RNF111 and RNF169 regulated by THs?

8. The authors conclude that expression of the five DEGs is enhanced in the wild type p53 background "because p53 inhibits the inhibitor (TH) of these genes and directly induces their expression". However, their expression levels in the p53WT/D2WT tumors seem only moderately higher than the levels in the p53KO/D2WT tumors, and substantially lower than the levels in the p53KO/D2KO tumors (Fig. 5g).

REVIEWER COMMENTS

Reviewer #1 - Thyroid hormone in cancer - (Remarks to the Author):

In this research work, Nappi et al, demonstrated that D2 is a novel p53-repressed target-gene in SCC. This conclusion is solidly based on several complementary methods including in vitro and in vivo data, which are a prerequisite for establishing/identifying genes as new p53-targets. The research work is comprehensive and systematic and the text is very well written, in a clear and organized manner. There are several major issues that need to be addressed/explained/discussed before considering publication at Nature Communications:

Major comments:

1. I agree that you have demonstrated that p53 suppresses D2 and thereby in p53 mutated SCC cells, it results in high (non-inhibited) D2 mRNA/protein levels and that this contributes to tumor progression. However, if this was the canonical pathway in all aggressive tumors, the majority of which are p53 mutated, we would expect high expression of D2 in all tumor types, as well as a correlation with worse survival. Analyses of the TCGA data, protein atlas and survival plots for D2, for an array of malignancies, clearly indicates that high D2 is not a general phenomenon.

For example in several tumors D2 is hardly expressed and in colon/endometrium high D2 is a "good prognostic factor". So D2 expression is highly heterogeneous and dynamic between tumors. What I suggest is to "de-generalize" the discovery. It should be clear in the text (abstract, introduction and discussion) that this association is established in SCC, but not in all tumors.

A. We thank the reviewer for highlighting this important point. We agree that the expression of D2 in different malignancies is highly variable. However, differently from the analysis cited by the reviewer, our analysis of TCGA data on the correlation between D2 expression and the percent of survival in different human malignancies demonstrated that in the majority of the tumor types high D2 expression correlates with a lower percent of survival. As indicated in the new figure S6, this is true in case of cervical cancer, liver cancer, lung, pancreatic, renal, stomach, thyroid and urothelial cancer. The discrepancies between our analysis and the reviewer analysis, as well as between different cancer types can be attributed to the variability of D2 expression during different phases of tumor progression (for instances, we observed that even in the same SCC tumor, D2 is poorly expressed in the early phase of tumorigenesis and raises at later stages).

To further address the link between p53-D2 and tumor malignancy, we also analyzed this correlation in TCGA Kaplan-Mayer plots observing a strong, inverse correlation between p53 and D2 that is already present at early stages of tumorigenesis, but with low significance, and gains much higher statistical significance at later stages of tumorigenesis, thus confirming our hypothesis (Figure S7-S9).

Thus, we conclude that although not in all the tumor types and not in the whole tumorigenic process, the overexpression of D2 can be considered a general mechanism fostering cancer progression in humans. We describe the new data in the Results (page 8) and in the Discussion (page 12), in which, as asked by the reviewer, we describe the high heterogeneity of D2 expression between different tumors and data in literature.

2. Based on your previous works, it seems that D2 acts opposite in BCC compared to SCC. In addition you demonstrated that the balance between D2 and D3 is different in both carcinomas. This should be discussed, as both models are presumably mutated in p53, but the outcome seems to be different.

A. We take the reviewer point, indeed, our published data (Di Girolamo et al., JCI 2015; Miro et al., Nature Communication 2020) as well as data in the present work indicate that while in poor invasive and most benign tumors like the BCC the TH signal is attenuated and the predominant deiodinase is D3, more invasive tumors with higher propensity to the metastatic process have enhanced TH signal and are characterized by D2 overexpression. The same process can be viewed in the context of the same tumor during the tumor progression, in which the hyperplastic and benign stages are characterized by high D3 expression, while evolution toward invasive stages is characterized by high D2 expression (Miro et al., Nature Communication 2020). So, even if BCC and SCC can undergo loss of p53, the final expression of D2 and D3 is different, suggesting the existence of other regulators of the D2/D3 balance besides p53.

We add these considerations in the Discussion (page 13).

3. You did not show D1 and D3 levels in the cell models/mice you used. This raises several questions: Did you search for p53-RE in D1 and D3?

Can there be a compensation mechanism by D1 when D2 is inhibited by WT p53? What happens to D3 in your samples- as you suggest that there is increased T3-bioavailability when p53 is mutated, is D3 repressed or not expressed at all in these SCC-manipulated cells?

A. In order to address this important question, we measured the expression levels of D1 and D3 in SCC cells following p53 overexpression. We searched for p53-binding sites in D1 and D3 promoter regions. As shown in the new figure S3a-b, in both the cases we found several putative p53 binding sites in the two promoters, thus suggesting that p53 can be viewed as a strong regulator of THs signaling through the regulation of the three deiodinases genes. In the case of SCC, such a regulation is only active on D2 and D3. Importantly, we also found that D3 expression is up-regulated by p53 and thus inversely correlates with D2 expression, indicating that D3 might be a positive p53 target gene (Figure S3c-d). We thank the reviewer that enabled us to add this very important finding that, once again, suggests that D2 and D3 are oppositely regulated in different settings. As far as for D1, we measured D1 mRNA expression and we found that D1 is undetectable in SCC cells and is also not expressed in p53-transfected cells.

We describe the new data in the Results (page 6) and in the Discussion (page 11-12).

4. One major issue is the fact that you refer to the SCC011 cells as "p53 WT". You cited ref #18 when referring to SCC011 as being p53 WT, but I could not find these specific cells mentioned in that specific reference. You further cite from ref. 59 "p53 wild-type SCC cell line (SCC011), previously described". In this ref. 59 it is written that the cells, which are also called "JHU0011", were given by Dr. J. Rocco. I searched and noticed that Rocco got the cells from David Sidransky, JH University (p63 mediates survival in squamous cell carcinoma by suppression of p73-dependent apoptosis. James W. Rocco et al, CANCER CELL 9, 45 13;56, 2006). These JHU0011/SCC011 cells are described at expasy: https://web.expasy.org/cellosaurus/CVCL_5986), as well as in earlier work by Sidranski. Anyhow, in all published works these cells are reported as p53 truncated mutants, resulting from a splice donor mutation following exon 6 and are therefore p53 null. I found another line which is also termed SCC011" and is from the same origin (Laryngeal SCC): https://web.expasy.org/cellosaurus/CVCL_7805 University of Turku-Squamous Cell Carcinoma-11, . These SCC011 cells are also p53 mutated, but with a deletion in AA187-196 in the DBD. This information is also nicely presented in figure 2 here: Genotyping and Characterization of HPV Status, Hypoxia, and Radiosensitivity in 22 Head and Neck Cancer Cell Lines, Göttingen et al., Cancers 2021, 13, 1069; <https://doi.org/10.3390/cancers13051069>). If the cells you used are from a different origin than the ones I found in the reverence you cited, than disregard my comments and provide the exact cell source, otherwise, this means that your results are based on mutated p53. The good news is that you used p53-transfected SCC011 cells in many

experiments in figure 1, so most results remain validated as you rescued the WT p53 and it inhibited D2. Anyhow, the exact cell line and its p53 status should be validated and if indeed mutated, corrected throughout the manuscript.

We thank the reviewer for raising this point and for providing the input to investigate in deeper detail the origin and p53 status of the SCC011 cells. We confirm that the cells we used in our experiments are the same described in James W. Rocco et al, CANCER CELL 9, 45 13;56, 2006. Consulting the "DEPMAP PORTAL (https://depmap.org/portal/cell_line/ACH-002249?tab=mutation), we also confirm that the reviewer is right, since this cell line, also called "JHU0011", is reported as p53 truncated mutant cell line, due to a splice donor mutation following exon 6 (annotation transcript ENST00000269305.4) that produces a p53 protein whose protein change is described as NA (Not Applicable). Thus, there are no proofs that the mutation attenuates or inhibits the p53 function. For this reason, we have assumed for our study that, compared to SCC13 cells carrying a C-T transition in codon 258 which inhibits the p53 function, JHU0011/SCC011 could behave like p53 wild-type cells. This assumption has been made also in light of the data we obtained that highlight a profound difference in p53 status between SCC13 and SCC011 (i.e. differential response to UV radiation, differential expression of p53 target genes etc.) all indicating that the basal p53 activity is much higher in SCC011 than in SCC13.

5. Figure 6: You use 30nM T3 concentrations which are considered in vitro supra-physiological by any measure (e.g Endocrinology, November 2015, 156(11):4325 13; 4335, doi: 10.1210/en.2015-1221). For near physiologic concentrations most groups use ~1nM T3. Unless you have indications showing that in your cell system under these conditions T3 levels are within the normal range, it should be clear that these are not physiological levels in the text.

A. We performed our studies in culture media containing 10% FBS, and endogenous hormone binding proteins present in the serum. Consequently, the effective free T3 concentration in the culture media -when we used 30 nM T3- was about 3nM, a dose that is ten times the physiological T3 concentrations in human plasma (1.2-3.4 nmol/L) but still compatible with the elevated T3 observed in hyperthyroid patients. Thus, we believe that the T3 concentrations used in our in vitro experiments can be considered a condition of mild T3 excess that overcome the physiological range.

By adding TH to the cell medium, you can actually activate the non-genomic $\alpha v\beta 3$ integrin axis. Unless you can validate that the cells are $\alpha v\beta 3$ negative, it cannot be fully ruled out that the results you received by adding exogenous T3 are non-genomic-mediated or D2-mediated. This possibility should be explained in the text and discussed.

A. To address the question about the activation of the non-genomic $\alpha v\beta 3$ integrin axis in our cell model, we measured the expression levels of pERK/ERK and p38MAPK in SCC011 and SCC13 cells treated with THs. As shown in the new Figure S4, we observed that the classical non-genomic targets of THs, namely pERK and p38MAPK are not changed by THs. Therefore, we conclude that in our experimental conditions, we are not affecting the non-genomic action of THs. These data are described in the Results (page 6).

The % of changes in the Annexin-PI assay are extremely low and only representative histograms are shown. The only panel that makes sense is panel d, but again, there is no guaranty that the results are mediated directly via the p53-D2 axis.

Therefore, the way I see it, the work presented in figure 6 weakens your very strong evidence presented in figures 1-5. I would omit it from this current work, unless you can substantiate the results based on the integrin expression and repeat the apoptosis assays to achieve more meaningful results.

A. We take the reviewer point and repeated the analysis of cellular apoptosis to strengthen our data. As shown in the new Figure 7, we found two very important and convincing results: (1) the percentage of apoptotic cells upon TH-treatment is slightly, but significantly increased compared to untreated cells; (2) when cultured in TH-deprived medium (charcoal stripped medium), Etoposide-induced apoptosis was dramatically reduced, thus reinforcing our initial hypothesis that TH is a potent promoter of apoptosis. Moreover, we verified that etoposide induces p53 and reduces D2 in our experiments by measuring D2 and p21 mRNA (Figure S16a). We describe these data in the Results (page 11).

Minor comments:

1. Manuscript title: I would mention D2 in the title, as it is the essence of this research work and it is important to convey the message that it is a novel p53-target gene. In addition, the oxidation part in the title is based on figure 6, which, as I mentioned in the comment above, demonstrates very limited results, which, in contrast to the remaining work, are not established as direct via the D2-p53 axis.

A. We agree with the reviewer and changed the title as suggested.

2. Figure 2: In Fig. 2e, the mutant p53 in SCC013 (E286G) is in the same AA site as one of the mutated p53 constructs you used in Fig. 2b-d (E286K). This might be a good addition for explaining the selection of these specific cells which carry an endogenous mutation at a similar DBD site. This also gives clinical relevance to your mutated construct and gives validation that the end of the DBD region is critical for p53 binding to the D2 promoter.

A. We thank the reviewer and agree with his/her considerations, which were added in the Discussion (page 11).

In Fig. 2g, although it is written that no effect was observed on D2 following UV irradiation in the p53-mutated cells, it is quite clear that there is a trend of elevation up to 8 hr. Maybe it did not reach statistical significance, but the trend is there and should be discussed or at least mentioned.

A. We thank the reviewer and added this discussion in the Results (page 5).

3. Figures 3 and 4: You generated heterozygous Trp53 gene deletion in the mice, but homozygous for D2. Does this mean that loss of one allele is sufficient to lose p53 activity towards D2 promoter (some sort of haploinsufficiency)? Please explain and if relevant discuss in the text.

A. Yes, our data suggest that even loss of one allele of p53 causes elevation of D2 expression. For instance, D2 expression in heterozygous p53 skin is already reduced when compared to wild type skin and is then totally reduced in p53^{-/-} skin. We discuss this new point in the Discussion (page 11-12).

Figure 3K and 3M. You show changes in several proteins by WB/IHC, respectively, but you do not show D2 itself. I understand that you provided the RNA level in the supplementary figure, but adding the protein level would add a lot to the figure validity (as you previously showed in Miro et al., 2019). Similarly in Figure 4G- D2 IHC will be a good addition.

A. We apologize with reviewer for not showing D2 protein levels in figures 3 and 4. Unfortunately, D2 antibodies are not available, even in commerce, and this does not allow the measurement of D2 protein by western blot or by IF. In our previous paper in Nature Communication, we used a Flagged-D2 knock in mouse model and the western blot were performed by using a Flag antibody. However,

since in this manuscript we measure the expression of different TH-target genes (Figure 3I, 3J, 3K, 5C-G, 6D) as well as the activation of a TH-report construct in Figure 1F, 1J, 2D, we believe that the data presented are indirect evidence that indicate that p53 down-regulates D2 at both, mRNA and protein levels.

4. Supplemental figures: Supp Figure 3A, right panel- what is D2-REC? it is not mentioned anywhere in the text and should be explained in the figure legend.

A. We now explain in the Figure Legend of the now-named Figure S5a that REC indicates the fragment generated by the recombination induced by the CRE expression, and thus results in D2 inactivation.

Supp 9b- The % of the HR reduction are minute. What is the meaning of such small changes?

A. The changes in figure S9b, now called S15c, although not potent, are statistically significant and indicate that the amount of repaired DNA is reduced to half of the total in the control, when cells are treated with TH. In our opinion, this is a substantial suggestion that TH is among the intracellular factors that reduce the ability to repair the DNA damage.

Reviewer #2 - SCC models - (Remarks to the Author):

This manuscript by Nappi and colleagues describes a mechanism for regulating levels of thyroid hormone activating enzyme type 2 deiodinase by p53. The authors cite a significant body of previous work indicating the T3 levels and hyperthyroidism are associated with increased incidence and more aggressive biology in some cancers.

These studies lead to the hypothesis that increased D2/TH signaling in cancer cells leads to tumor progression and aggressive biology. Therefore what mechanism is responsible for increased D2/TH signaling in cancers.

To evaluate this hypothesis the investigators conducted an in silico analysis and identified potential binding sites for p53 in the Dio2 promoter region which was supported by an inverse correlation between TP53 and ATM and DIO2 expression. The range of assays used including a combination of biochemical assays on cell lines and genetic models confirm that p53 binds to a predicted P1 sequence in the promoter to suppress D2 mRNA production.

Although much of the data appear reasonable there is a major experimental design flaw that negatively impacts the relevance of these data to understanding UV-induced skin cancer. The manuscript is well written.

Major Issue: The UV irradiation is performed using a Stratlinker. Stratlinkers emit predominantly UVC which is not a relevant cause of UV-induced skin cancer because it is absorbed by the atmosphere and does not reach the earth's surface. These experiments should be done using a mixture of UVB and UVA emitting lamps (there are many available). UVC filters should be in place to eliminate any UVC that may be emitted.

A. We thank the reviewer for highlighting this important point. Our data using the Stratlinker as UV lamp can only indicate that in case of p53 inactivation, D2 expression is up-regulated. However, this does not represent the real condition of human exposure to UV-A and B radiations. Thus, to

strengthen our important conclusions, as suggested by the reviewer, we repeated some experiments of UV radiation by using a different UV lamp, namely, the six Philips TL12/60W fluorescent lamps. As shown in the new figure 2i-l, also UV-A and UV-B radiation generated both, p53 phosphorylation and D2 suppression in SCC11 cells while did not change the p53-D2 axis in SCC13 cells.

Other issues mentioned below should be addressed.

1. The authors tend to overstate the significance of Dio2 in p53-regulated tumorigenesis. Though the data are intriguing, they show that Dio2 is a downstream effector of p53 and likely a key regulator of metabolism in neoplastic cells.

Downregulation of Dio2 is dependent on p53, the authors infer that DNA damage is regulated by Dio2 levels. This is speculative because they have not shown that this is independent of p53 loss of function.

A. In order to strengthen our conclusion that D2 is responsible for the modulation of genes involved in the DNA repair independently of p53 status, we performed experiments in SCC13 cells that stably carry an inducible Tet-ON-D2 construct, generated in our lab (Miro et al., *Cancers* 2021). In these cells, Doxycycline administration turns on the D2 expression. Moreover, the cells are p53 null. As shown in the new Figure 6c-d, the DOX-induced D2 increased the 8-oxo-dG and γ -H2AX/RAD51 staining. Moreover, the DOX-induced D2 significantly reduced the expression of the DNA-repair genes (Fig. S11e).

Furthermore, a similar result can be seen in Figure S11d in which D2 depletion enhances the expression of the DEGs and T3 treatment rescues this effect. These data are now described in the Results (page 11).

Sentences addressing this in lines 354-364 should be revised, note that DSBs typically do not occur during UV-treatment of keratinocytes. NER and SS DNA breaks occur. Not clear that TH induced DSBs are relevant to UV induced skin cancer biology.

A. We agree with the reviewer that UV exposure induces primarily NER and SS, however, prolonged and intense UV radiation culminates also in accumulating DSB (Greiner R. et al., *Nucleic Acid Res* 2012; Federico MB et al., *PlosGenetics* 2016). In our experimental conditions, the generation of DSB following UV radiation is demonstrated by the positive staining of cells with γ -H2AX antibody (Figure 6c), thus reinforcing the concept that TH treatment augments the DNA damage and DSBs.

2. The authors focus on potential increased metastasis as a result of disrupted p53/D2 signaling which is important. However, they show this pathway is disrupted in SCCIS and since SCCIS have been known to have classic oncogenic p53 mutations in approx. 60% of lesions (PMID 32649944) the biological impact of disrupted p53/D2 signaling would be seen in a majority of these lesions. This should be discussed as SCCIS are much more common than cSCCs.

A. We take the reviewer point and added a new statement in the Discussion section in which we speculate that the mechanism of p53-D2-TH regulation can be extended to many different malignancies in which p53 is loss or mutated. Indeed, analysis of a TCGA data set indicate that D2 expression can be seen in different malignancies, and that D2 high expression correlates with lower percent of survival (Fig. S6-S9).

3. The 8oxodG staining technique should be clarified, were tissue sections protected from atmospheric oxygen which promotes 8oxodG formation? Also in the control panels in Fig 6a, the 8oxodG appears not to overlap with the DAPI stain. Is that all mitochondrial DNA staining? If the data is to be shown there should be some type of quantitative analysis.

A. We did not protect the tissues from atmospheric oxygen for both, control and TH-treated cells, but the background of 8-oxo-dG in CTR cells was very low in our experimental conditions.

Fig 6a shows the DNA oxidation induced by TH measured by confocal microscopy. We have been very careful in assessing the reproducibility of the co-localization signals by running several replicates (3-4 for each point). TH treatment induced a statistically significant accumulation of nuclear 8-oxo-dG. The perinuclear signal does not overlap with DAPI and is likely the mitochondria (10.1016/S0891-5849(00)00185-4). We measured the co-localization with DAPI as reported in the new Fig.S15a. As also noted by the reviewer, although there are DAPI-nuclear DNA signals, we find the majority does not overlap. This can be explained by the concept that THs induce the accumulation of oxidized nucleotides in mitochondrial DNA (Thyroid Res 5, 24, 2012).

Reviewer #3 - p53 - (Remarks to the Author):

This study shows that the tumor suppressor p53 downregulates expression of type 2 Deiodinase (D2) which catalyzes conversion of thyroid hormone T4 to T3 and thereby increases availability of active thyroid hormone. The authors demonstrate that p53 inactivation leads to elevated D2 expression and increased fitness of tumor cells via regulation of genes involved in DNA damage and repair and redox signaling. This suggests that D2 or thyroid hormone could be relevant targets for treatment of TP53 mutant tumors. The data presented in this study provide a novel understanding of the impact of TP53 mutations in tumor progression and clearly have very interesting implications for novel cancer therapy.

Specific comments:

1. Overall, the data are solid and support the author's conclusions. However, one problem is that p53-mediated suppression of transcription is still poorly understood at the molecular level and highly controversial. While it is well established that p53 binding to DNA motifs is essential for p53-mediated transcriptional transactivation of classical target genes such as p21, Bax and Puma, the molecular basis for p53-mediated suppression of transcription has not been completely elucidated. The authors show that D2 has two p53 binding motifs in its promoter and that P1 is critical for suppression. Yet it is unclear why p53 binding to the D2 promoter leads to suppression of transcription whereas p53 binding to for instance the p21 promoter is known to activate transcription. The authors should discuss this point.

A. We agree with the reviewer that p53 acts mainly as transcriptional activator, while its role in transcriptional repression is less known. However, different reports also indicate a role of p53 in transcription inhibition (as in the case reported by Tatiana Grohmann Ortolan and Carlos Frederico M. Menck, PLOS One 2013). Also in our manuscript, the new data in the revised version show that p53 down-regulates D2 while positively regulating D3 (Fig. S3). Our speculation is that in different promoter regions, p53 can combine with co-activators or co-repressors, thus exerting opposite effects on overall gene transcription.

2. In agreement with the general concept of p53-dependent transactivation of transcription, mRNA levels for p53 target genes p21 and Bax are induced upon UV treatment in TP53 wild type SCC cells (Suppl Fig. 2a). However, it is surprising that the canonical p53 target gene MDM2 is strongly suppressed. Could the authors provide a plausible explanation?

A. We thank the reviewer for highlighting this point that is indeed complicated. We agree that the drastic loss of MDM2 is somehow unexpected. However, some report has suggested that, upon p53 activation, MDM2 can mediate its own down-regulation in a feedback loop mechanism (Lu X et al., 2007, Cancer Cell; Toledo F, 2006 Nat Rev Cancer), thus providing a rationale for the reduction of MDM2 in UV-radiation experiments.

3. The authors have examined D2 expression in the presence and absence of functional p53 in UV-treated SCC011 (wt p53) and SCC13 (mut p53) cells in Fig. 2. However, the exact role of D2 and TH in the context of a p53-mediated cellular stress response in normal cells has not been examined. Does activation of wt p53 by DNA damage in normal cells affect D2 and DEGs in a similar way as shown for the SCC cells? This could be tested in for example human diploid fibroblasts or primary keratinocytes.

A. We take the reviewer point and studied the regulation of D2 by p53 in a context of non-tumoral cells as the human skin-derived primary fibroblasts and in HaCaT cells (immortalized human keratinocytes). In the case of fibroblasts, we did not observe any regulation, probably because the basal expression level of D2 is very low in these cells. Interestingly, in HaCaT cells p53 overexpression caused D2 mRNA inhibition, similarly to what observed in SCC cells. Thus, we conclude that p53 negatively regulates D2 in different pathophysiological conditions. These data are shown in the new Fig. S2a and S14b and described in Results (pages 5 and 9) and in the Discussion (page 13).

4. The defective DNA repair and enhanced DNA damage in cells with elevated THs is intriguing but no data on p53 and D2 are shown (Fig. 6). Is p53 induced by etoposide in the SCC011 cells? Is D2 downregulated? SCC13 cells with mutant p53 should be tested in parallel for comparison.

A. Yes, we controlled D2 regulation and p53 activation by measuring D2 and p21 mRNA as shown in Fig. S16a, which indirectly confirmed that p53 is activated upon etoposide treatment. Similarly, D2 is reduced. To address the second point about the effect of TH on Etoposide-induced DNA damage, as asked by the reviewer, we performed the experiment in SCC13 cells. As shown in the new Fig. 7c, TH increased the percentage of necrotic cells with and without Etoposide.

5. According to Fig. 2h, p53 levels are high even in the untreated SCC011 cells, there is no significant further accumulation of p53 upon UV irradiation, although p53 Ser-15 phosphorylation is increased at 4 and 8 hours. Could the authors comment on this?

A. As stated by the reviewer, in our experimental conditions, the UV-treatment induces only the phosphorylation of p53 in Ser-15 and does not induce accumulation of the wild type p53. However, this is still an indication of p53 activation and is in agreement with many other data in literature reporting that p53 activation can be due to both protein accumulation and phosphorylation or only to the increased phosphorylation. For instance, a very recent article (Annika Wylie et al., Developmental Cell 2022) explains how, in *Drosophila melanogaster*, p53 activation is mostly based on its phosphorylation in Ser-15.

6. Suppl Fig. 5e needs some clarification. The figure shows reduced expression levels of the five top DEGs in tumors from mice with THs in the drinking water. Does the figure show ratio between expression levels in the D2KO and D2WT tumors with and without THs?

A. We thank the reviewer, this is an error. We mislabeled the Figure since the experiment has been performed in wild type animals treated or not with THs in drinking water. In this experiment the

down regulation of the DEGs is a further confirm that THs are up-stream regulators of their expression. We corrected the Figure Legend and the Figure, now Fig. S11f.

7. The five identified DEGs all have multiple TREs (Suppl Figs. 6-8). Yet according to the ChIP data in Fig. 5e TRs bind physically only to MAD1L1 and POLD3. What is the reason for the differential binding despite the presence of TREs? How are PAK1, RNF111 and RNF169 regulated by THs?

A. The reviewer is right since, although all the identified DEGs show multiple putative TREs, only two of them (MAD1L1 and POLD3) are physically bound by TR in our ChIP analysis. We argue that the other DEGs can be considered TH-target genes based on the RNAseq data, but could either be indirect targets of THs or (having a TRE) could be target of TH in different cellular contexts and not in SCC cells.

8. The authors conclude that expression of the five DEGs is enhanced in the wild type p53 background "because p53 inhibits the inhibitor (TH) of these genes and directly induces their expression". However, their expression levels in the p53WT/D2WT tumors seem only moderately higher than the levels in the p53KO/D2WT tumors, and substantially lower than the levels in the p53KO/D2KO tumors (Fig. 5g).

A. We agree with the reviewer and explain the mild effects on DEGs modulation due to the sample analyzed in Fig. 5g which is the total skin (including epidermis and derma) from p53WT;D2WT, p53KO;D2WT and p53KO;D2KO mice. We believe that measuring the levels of the DEGs in *in vivo* context dilute the overall effects of p53-dependent and TH-dependent regulation since in this case we have a pull of different cell types (namely, keratinocytes, fibroblasts, immune cells and many others), while the effects seen in SCC cells are much more pronounced.

Reviewers' Comments:

Reviewer #1:

Remarks to the Author:

The authors have addressed all of my concerns. My only and last minor comment is that SCC011 cells cannot be still attributed to simply as "p53 WT" activity. In the DEPMAP portal it is clearly written that the mutation in p53 in SCC011 cells is under "damaging" annotation- meaning that the protein activity/function is altered:

TP53 17 7578175 7578175 Splice_Site damaging SNP

This "damaging" annotation may lead to either loss-of-function or gain-of-function activities. As it is widely known that each p53 mutant may "play a different tune" on its target genes, I therefore still recommend not to refer to the p53 status as WT (because they are mutated and it is confusing), rather to propose that this mutated form has not lost its ability to bind the D2 promoter (and other known target genes as you clearly present in Supp Fig. 2b) and exert its activity, which is either repress/activate the target genes.

Reviewer #2:

Remarks to the Author:

The authors have addressed most of my concerns. The relevance of these pathways to UV-induced skin cancer is still unclear; the mention of UV-induced skin cancer should be de-emphasized and the use of UVC should be described as a tool for inducing DNA damage with a substantial proportion of DS breaks not as a model for UV-induced skin cancer.

Reviewer #3:

Remarks to the Author:

The authors have adequately addressed several of my specific points. However, I would like to comment on their responses to points 3, 4 and 6:

3. The authors have tested the effect of p53 on D2 and DEGs in HaCaT cells by transfection of a p53 expression plasmid. Suppression of D2 (Fig. S2a) and upregulation of DEGs (Fig. S14b) was observed. This is fine but does not fully address my question which was whether activation of p53 by DNA damage would affect D2 and DEGs in normal cells in the same way as in the SCC cells. Did the authors examine whether induction of endogenous p53 by UV in HaCaT cells leads to downregulation of D2?

4. Fig. S16a does not include data on p53 but shows that D2 mRNA is induced (moderately) and that p21 mRNA is more or less unaffected upon treatment with etoposide. This does not confirm that p53 is activated by etoposide in these cells, rather the opposite.

6. My question was whether the figure shows ratios between expression levels in the D2KO and D2WT tumors with and without TH? What is shown on the y axis?

Point-by-point to the reviewers' comments

Reviewer #1 (Remarks to the Author):

The authors have addressed all of my concerns. My only and last minor comment is that SCC011 cells cannot be still attributed to simply as "p53 WT" activity. In the DEPMAP portal it is clearly written that the mutation in p53 in SCC011 cells is under "damaging" annotation- meaning that the protein activity/function is altered: TP53 17 7578175 7578175 Splice_Site damaging SNP. This 'damaging' annotation may lead to either loss-of-function or gain-of-function activities. As it is widely known that each p53 mutant may "play a different tune" on its target genes, I therefore still recommend not to refer to the p53 status as WT (because they are mutated and it is confusing), rather to propose that this mutated form has not lost its ability to bind the D2 promoter (and other known target genes as you clearly present in Supp Fig. 2b) and exert its activity, which is either repress/activate the target genes.

A. As requested by the reviewer, we changed the annotation WT for SCC011 cells in the manuscript and in the figures. Moreover, we specify in the Materials and Methods that this cell line is reported having a "damaging" annotation which may lead to either loss-of-function or gain-of-function activities, and that in our experiments these cells behave as cells with a functional p53, at least for the D2 and p21 regulation".

Reviewer #2 (Remarks to the Author):

The authors have addressed most of my concerns. The relevance of these pathways to UV-induced skin cancer is still unclear; the mention of UV-induced skin cancer should be de-emphasized and the use of UVC should be described as a tool for inducing DNA damage with a substantial proportion of DS breaks not as a model for UV-induced skin cancer.

A. We thank the reviewers and have indicated in the text that "We performed UV radiation experiments as a tool for inducing DNA damage with a substantial proportion of DS breaks and p53 activation". See Discussion, page 12.

Reviewer #3 (Remarks to the Author):

The authors have adequately addressed several of my specific points. However, I would like to comment on their responses to points 3, 4 and 6:

3. The authors have tested the effect of p53 on D2 and DEGs in HaCaT cells by transfection of a p53 expression plasmid. Suppression of D2 (Fig. S2a) and upregulation of DEGs (Fig. S14b) was observed. This is fine but does not fully address my question which was whether activation of p53 by DNA damage would affect D2 and DEGs in normal cells in the same way as in the SCC cells. Did the authors examine whether induction of endogenous p53 by UV in HaCaT cells leads to downregulation of D2?

A. We performed a new experiment in HaCaT cells exposed to UV radiation. Also in this case we observed that UV exposure induces p53 activation (indirectly measured as the induction of p21 mRNA) and D2 mRNA down-regulation (new Supplementary Figure 2d).

4. Fig. S16a does not include data on p53 but shows that D2 mRNA is induced (moderately) and that p21 mRNA is more or less unaffected upon treatment with etoposide. This does not confirm that p53 is activated by etoposide in these cells, rather the opposite.

A. We apologize for the mistake. We made an error in the Figure, indicating that cells were treated with Etoposide and Kudos. In the Figure legend we did not make the same mistake and is correctly indicated that cells were treated with THs and Etoposide. In detail, we show that in the experiments addressing the THs-induced apoptosis, Etoposide reduces D2 expression and enhances p21 and that THs partially attenuate these effects.

6. *My question was whether the figure shows ratios between expression levels in the D2KO and D2WT tumors with and without TH? What is shown on the y axis?*

A. The figure does not show ratios between expression levels in the D2KO and D2WT tumors with and without THs, but the Log₂ Gene Expression of *Mad11l*, *Pak1*, *Pold3*, *Rnf111* and *Rnf169* genes in CTR mice, treated or not with thyroid hormones in drinking water. We also corrected the figure by adding the information “Log₂ Gene Expression” on the y axis.